# A Geostatistical Approach to Estimate High Resolution Nocturnal Bird Migration Densities from a Weather Radar Network

**Raphaël Nussbaumer** [1,2,*], **Lionel Benoit** [2], **Grégoire Mariethoz** [2], **Felix Liechti** [1], **Silke Bauer** [1] **and Baptiste Schmid** [1]

1   Swiss Ornithological Institute, 6204 Sempach, Switzerland; Felix.Liechti@vogelwarte.ch (F.L.); Silke.Bauer@vogelwarte.ch (S.B.); Baptiste.Schmid@vogelwarte.ch (B.S.)
2   Institute of Earth Surface Dynamics, University of Lausanne, 1015 Lausanne, Switzerland; Lionel.Benoit@unil.ch (L.B.); Gregoire.Mariethoz@unil.ch (G.M.)
*   Correspondence: raphael.nussbaumer@vogelwarte.ch

**Abstract:** Quantifying nocturnal bird migration at high resolution is essential for (1) understanding the phenology of migration and its drivers, (2) identifying critical spatio-temporal protection zones for migratory birds, and (3) assessing the risk of collision with artificial structures. We propose a tailored geostatistical model to interpolate migration intensity monitored by a network of weather radars. The model is applied to data collected in autumn 2016 from 69 European weather radars. To validate the model, we performed a cross-validation and also compared our interpolation results with independent measurements of two bird radars. Our model estimated bird densities at high resolution (0.2° latitude–longitude, 15 min) and assessed the associated uncertainty. Within the area covered by the radar network, we estimated that around 120 million birds were simultaneously in flight (10–90 quantiles: 107–134). Local estimations can be easily visualized and retrieved from a dedicated interactive website. This proof-of-concept study demonstrates that a network of weather radar is able to quantify bird migration at high resolution and accuracy. The model presented has the ability to monitor population of migratory birds at scales ranging from regional to continental in space and daily to yearly in time. Near-real-time estimation should soon be possible with an update of the infrastructure and processing software.

**Keywords:** aeroecology; bird migration; geostatistical modeling; interactive visualization; kriging; radar network; spatio-temporal interpolation map; weather radar

## 1. Introduction

Every year, several billions of birds undergo migratory journeys between their breeding and non-breeding grounds [1,2]. These migratory movements link ecosystems and biodiversity on a global scale [3], and their understanding and protection require international efforts [4]. Indeed, declines in many migratory bird populations [5,6] resulted from the rapid changes in their habitats, including the aerosphere [7]. Changes in aerial habitats are diverse, and their consequences still poorly known. Climate change may alter global wind patterns and consequently the wind assistance provided to migrants [8]. Likely to be more severe, the impact of direct anthropogenic changes, including light pollution that reroutes migrants [9], buildings [10], wind energy production [11], and aviation [12], causes billions of fatalities every year [13].

In the face of these threats and to set up efficient management actions, we need to quantify bird migration at various spatial and temporal scales. Fine-scale monitoring is crucial for understanding the phenology of migration and its drivers, identifying critical spatio-temporal protection zones to

support conservation actions, and assessing collision risks with artificial structures and aviation to inform stakeholders. However, the great majority of migratory landbirds fly at night [14], rendering the quantification of the sheer scale of bird migration a challenging exercise.

Radar monitoring has the potential to quantify birds' migratory movements at the continental scale [15]. Initially limited to single dedicated short-range radars, radar aeroecology truly took off when it was able to leverage existing weather radar networks, thus providing continuous monitoring over large geographical areas such as Europe or North America [2,16–19]. One important challenge in using networks of weather radars is the interpolation of their signals in space and time. Recent studies [2,19] have used relatively simple interpolation methods as they targeted patterns at coarse spatial and temporal scales. However, these methods are insufficient if higher spatial or temporal resolution is needed, such as for the fundamental and applied challenges outlined above.

To achieve a high-resolution interpolation of migration intensity derived from weather radars ( 20 km–15 min), we propose a tailored geostatistical framework able to model the spatio-temporal patterns of bird migration. Starting from time series of bird densities measured by a radar network, our geostatistical model produces a continuous map of bird densities over time and space. A major strength of this method is its ability to provide the full range of uncertainty and thus to evaluate complex statistics, for instance, the probability that bird densities reach a given threshold. In addition to the estimation map, the method also produces simulation maps which are essential for several applications such as quantification of the total number of birds.

As a proof-of-concept, we applied our geostatistical model to a three-weeks dataset from the European Network of weather radars [20] and validated the results with independent dedicated bird radars. In addition to insights into the spatio-temporal scales of broad-front migration, our approach provides high-resolution (0.2° latitude and longitude, 15 min) interactive maps of the densities of migratory birds.

## 2. Materials and Methods

### 2.1. Weather Radar Dataset

Our dataset originates from measurements of 69 European weather radars, spread from Finland to the Pyrenees (eight countries) and covering the period from 19 September to 10 October 2016 (Figure 1). It thus encompasses a large part of the Western European flyway during fall migration 2016.

Based on the reflectivity measurements of these weather radars, we used the bird densities as calculated and stored on the repository of the European Network for the Radar surveillance of Animal Movement (ENRAM) (https://github.com/enram/data-repository) ([21] for details on the conversion procedure). We inspected the vertical profiles and manually cleaned the bird densities data (see detailed procedure in Appendix A and resulting vertical profiles in Supplementary Material S1).

As we targeted a 2D model, we vertically integrated the cleaned bird densities from the radar elevation to 5000 m above sea level. Because we aimed at quantifying nocturnal migration, we restricted our data to night-time, between local dusk and dawn (civil twilight, sun 6° below horizon). Furthermore, as rain could contaminate and distort the bird densities calculated from radar data, a mask for rain was created when the total column of rain water exceeds a threshold of 1 mm/h (ERA5 dataset from [22]). In the end, the resulting dataset consisted of a time series of nocturnal bird densities [bird/km$^2$] at each radar site with a resolution of 15 min (Figure 2).

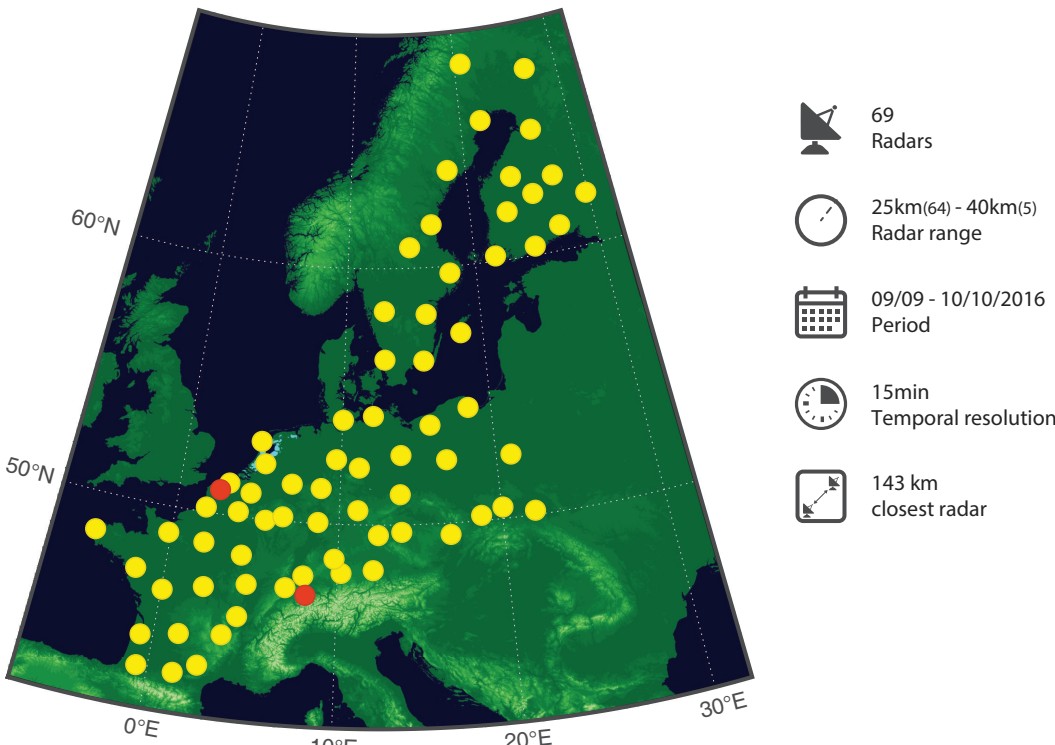

**Figure 1.** (**Left**) Locations of weather radars of the European Network for the Radar surveillance of Animal Movement (ENRAM) network, whose fall 2016-data were used in this study (yellow dots) and the two dedicated bird radars for validation (red dots). (**Right**) Key characteristics of the dataset used.

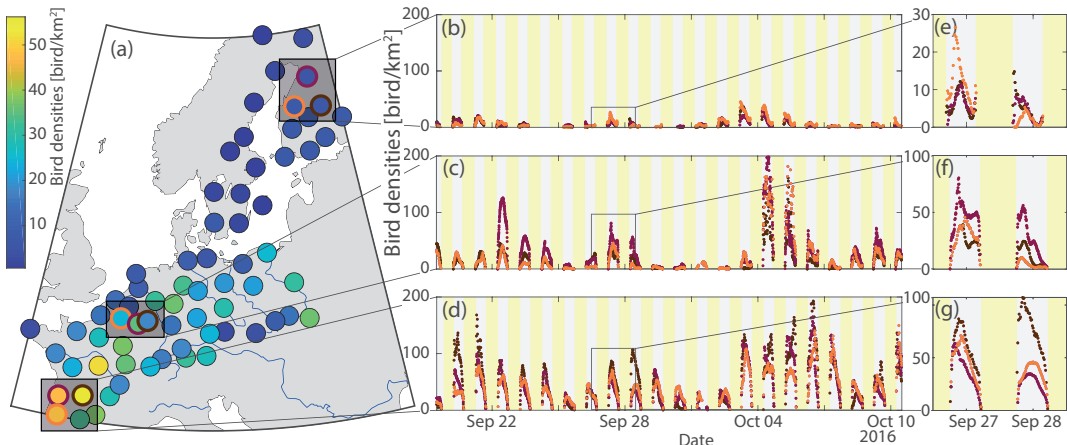

**Figure 2.** Illustration of the spatio-temporal variability of bird densities measured by a weather radar network. (**a**) Average bird densities measured by each radar over the whole study period. (**b**–**d**) Time series of bird densities measured by the radar with the corresponding outer ring color in panel (**a**). (**e**–**g**) Zoom on a two-days period. A strong continental trend appears in panel (**a**) as well as a correlation at the multi-night scale when comparing (**b**–**d**). These spatial correlations are even stronger at the regional scale when comparing within a subplot (**f**). The intra-night scale shows an obvious bell-shape curve pattern during each night (**g**).

## 2.2. Interpolation Approach

Bird densities are strongly correlated spatially (Figure 2a) and temporally at both nightly (Figure 2b–d) and sub-nightly scales (Figure 2e–g). These strong spatio-temporal correlations motivated the use of a Gaussian process regression to interpolate bird densities measured by weather radars at high temporal resolution. In this framework, the spatio-temporal structure of bird migration is first

learned from the punctual radar measurements. This model is then combined with the measurements to estimate bird densities at any location in space and time. In this paper, we adopt the terminology and notations of Geostatistics [23,24], and mention its correspondence with Gaussian processes in the field in machine learning [25].

Because of the multi-scale temporal structure of bird migration (Figure 2), we consider here an additive model combining two temporal scales: first, a multi-night process that models bird density averaged over the night and, second, an intra-night process that models variations within each night. Subsequently, each scale-specific process is further split into two terms: a smoothly-varying (in space time) deterministic trend and a stationary Gaussian process.

### 2.3. Geostatistical Model

The bird density $B(\mathbf{s}, t)$ observed at location $\mathbf{s}$ and at time $t$ is modeled by

$$B(\mathbf{s}, t)^p = \underbrace{\mu(\mathbf{s}) + M(\mathbf{s}, d(t))}_{\text{Multi-night scale}} + \underbrace{\iota(\mathbf{s}, t) + I(\mathbf{s}, t)}_{\text{Intra-night scale}}, \tag{1}$$

where $\mu$ and $\iota$ are deterministic trends at the multi-night and intra-night resolution, respectively, and $M$ and $I$ are random effects at the multi-night and intra-night resolution, respectively; $d(t)$ is a step function that maps the continuous time $t$ to the discrete day $d$ of the closest night. A power transformation $p$ is applied on bird densities to transform the highly skewed marginal distribution into a Gaussian distribution (Figure A2 in Appendix B). Figure 3 illustrates the decomposition of Equation (1) during three nights.

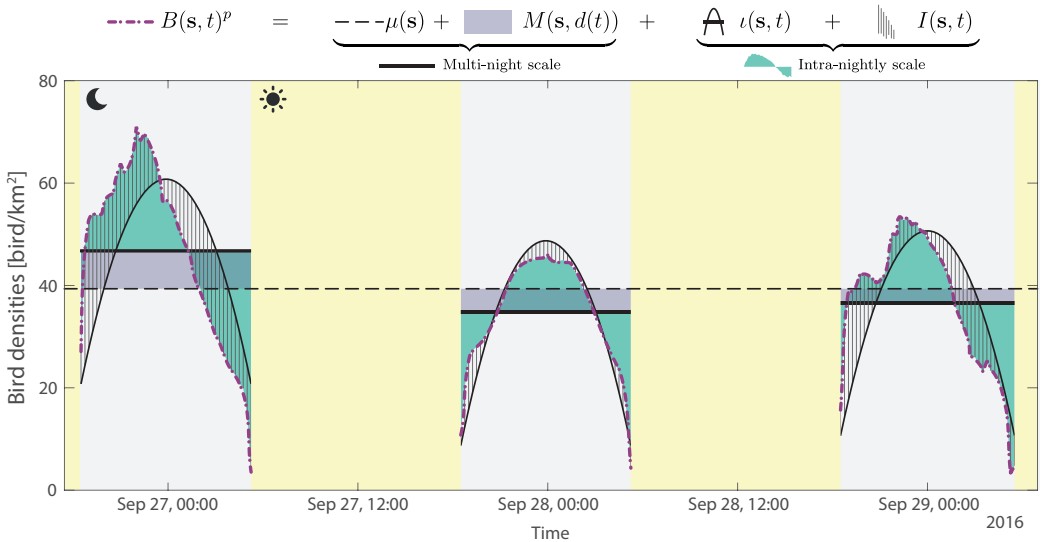

**Figure 3.** Illustration of the proposed mathematical model decomposition of Equation (1) with the exception that the power transformation was not applied. Note that the values of $M$, $\iota$, and $I$ can be either positive or negative.

#### 2.3.1. Multi-Night Scale

At the multi-night scale, we model bird densities averaged overnight as a space-time Gaussian process with a spatial trend (also called mean function), due to the general increasing bird densities southwards (Figure 2a). Because of the relatively short duration of the dataset, no temporal component is added in the trend. The trend at the multi-night scale is therefore modeled as a planar function,

$$\mu\left(\mathbf{s} = [s_{\text{lat}}, s_{\text{lon}}]\right) = w_{\text{lat}} s_{\text{lat}} + w_0, \tag{2}$$

where $s_{\text{lat}}$ and $s_{\text{lon}}$ are the latitude and longitude of location **s** ; $w_{\text{lat}}$ is the slope coefficient in latitude; and $w_0$ is the value of the trend at the origin. Because no longitudinal trend is observed in the data (Figure 1a), only latitude is used to configure the planar function (see Figure A3 in Appendix B). If longer periods are considered, Equation (2) can be replaced by a spatio-temporal polynomial function in order to handle the emerging patterns of long-term nonstationarity.

With the spatial trend accounted for by $\mu$, $M$ can be modeled as a Gaussian random process with zero-mean. $M$ is assumed to be 2nd order stationary, so that its covariance function $C_M$ (also called autocovariance or kernel function) depends only on $\Delta \mathbf{s}, \Delta t$,

$$C_M\left(M(\mathbf{s},t), M(\mathbf{s}+\Delta\mathbf{s}, t+\Delta t)\right) = C_M\left(\Delta\mathbf{s}, \Delta t\right). \tag{3}$$

The covariance function $C_M$ is modeled with the Gneiting type function [26]

$$C_M\left(\Delta\mathbf{s}, \Delta t\right) = C_0\delta_{\Delta\mathbf{s}} + \frac{C_G}{(\Delta t/r_t)^{2\delta}+1}\exp\left(\frac{-\left(\|\Delta\mathbf{s}\|/r_s\right)^{2\gamma}}{\left(\left(\Delta t/r_t\right)^{2\alpha}+1\right)^{\beta\gamma}}\right). \tag{4}$$

In Equation (4), the scale parameters $r_t$ and $r_s$ (in space and time, respectively) control the decorrelation distances and, thus, the average extent and duration of the space-time patterns of $M$. The regularity parameters $0 < \alpha$ and $\gamma < 1$ (in space and time, respectively) control the shape of the covariance function close to the origin. Values of $\alpha$ and $\gamma$ close to 0 lead to sharp variations at short lags, whereas values close to 1 lead to smooth variations of $M$. The separability parameter $\beta$ controls the space-time interactions. When $\beta = 0$ the space-time interactions vanish and the covariance function becomes space-time separable. $C_G$ controls the amplitude of the covariance function. Finally, $C_0$ is a nugget effect which accounts for the uncorrelated variability of $M$, and $\delta$ is the Kronecker delta function

$$C_0\delta_{\Delta\mathbf{s}} = \begin{cases} C_0 & \text{if } \Delta\mathbf{s} = \mathbf{0} \\ 0 & \text{otherwise} \end{cases}. \tag{5}$$

Note that in contrast to a usual nugget $C_0\delta_{\Delta\mathbf{s},\delta t}$, here we use a nugget in the space dimension only $\delta_{\Delta\mathbf{s}}$. This nugget accounts for the uncorrelated variability of $M$ over space which can be caused by persistent local geographical features affecting bird migration (e.g., topography and water body) or possible bias of radar observation (e.g., ground scattering). The total variance of $M$ is defined as $C_M(0,0) = C_0 + C_G$.

### 2.3.2. Intra-Night Scale

At the intra-night scale, the main trend visible in the dataset is a bell-shape curve pattern (Figure 2e–g) that results from the onset and sharp increase of migration activity after sunset, and its slow decrease towards sunrise [27]. This trend is modeled with a curve template $\iota$ for all nights and locations, defined by a polynomial of degree 8,

$$\iota(\mathbf{s},t) = \sum_{i=0}^{i=8} a_i NNT(\mathbf{s},t)^i, \tag{6}$$

where $a_i$ are the coefficients of the polynomial and $NNT$ (Normalized Night Time) is a proxy of the progression of night, defined such that the local sunrise and sunset occur at $NNT = -1$ and $NNT = 1$, respectively.

$$\tilde{I}(\mathbf{s},t) = \frac{I(\mathbf{s},t)}{\sigma_I(\mathbf{s},t)}, \tag{7}$$

where $\sigma_I(\mathbf{s}, t)$ is a polynomial function with coefficients $b_i$ that models the variation of the variance

$$\sigma_I(\mathbf{s}, t) = \sum_{i=0}^{i=10} b_i NNT(\mathbf{s}, t)^i. \tag{8}$$

This normalization allows to use a stationary covariance function for $\tilde{I}$,

$$C_{\tilde{I}}(\Delta \mathbf{s}, \Delta t) = C_0 \delta_{\Delta \mathbf{s}, \Delta t} + \frac{C_G}{(\Delta t / r_t)^{2\alpha} + 1} \exp\left(\frac{-\left(\|\Delta \mathbf{s}\| / r_s\right)^{2\gamma}}{\left((\Delta t / r_t)^{2\alpha} + 1\right)^{\beta\gamma}}\right). \tag{9}$$

Note that modeling $I$ through the covariance function of its normalized variable $\tilde{I}$ is equivalent to modeling $I$ directly with a nonstationary covariance function, which includes $\sigma_I(\mathbf{s}, t)$.

### 2.4. Bird Migration Mapping

The geostatistical model presented in Section 2.3 is used to interpolate bird density observations derived from weather radars, and produces high resolution maps of both estimation (with corresponding uncertainty) and simulation (see Section 2.4.2). In the case study presented in this paper, the interpolation map is calculated on a spatio-temporal grid with a resolution of $0.2°$ in latitude ($43°$ to $68°$) and longitude ($-5°$ to $30°$) and 15 min in time, resulting in $127 \times 176 \times 2017$ nodes. Over this large data cube, the estimation and simulation are only computed at the nodes located (1) over land, (2) within 200km of the nearest radar and (3) during night-time ($-1 < NNT(t, \mathbf{s}) < 1$).

#### 2.4.1. Estimation

The estimation is performed by applying universal kriging at both the multi- and intra-night scales. We employ a two-step approach of universal kriging where the trends $\mu$ and $\iota$ are first estimated by ordinary least squares (see Appendix B.2 and B.3), and then subtracted from the observations. The resulting random effects $M$ and $\tilde{I}$ are parameterized by fitting their covariance function (defined in Equations (4) and (9)) to the empirical covariance computed based on the detrended observations (see Appendix B.4). Finally, $M$ and $\tilde{I}$ are interpolated at any space-time location of interest, $\mathbf{s}_0, t_0$, by simple kriging (see Appendix C), denoted as $M(\mathbf{s}_0, t_0)^*$ and $\tilde{I}(\mathbf{s}_0, t_0)^*$. The final estimation of bird density $B^p(\mathbf{s}_0, t_0)^*$ is reconstructed based on Equation (1) as,

$$B^p(\mathbf{s}_0, t_0)^* = t(\mathbf{s}_0) + M(\mathbf{s}_0, t_0)^* + \iota(t_0) + \sigma_I(\mathbf{s}_0, t_0) \tilde{I}(\mathbf{s}_0, t_0)^*. \tag{10}$$

An important advantage of using kriging is that it expresses the estimation as a Gaussian distribution, thus providing not only the "most likely value" (i.e., mean or expected value), but also a measure of uncertainty with the variance of estimation. As $M$ and $\tilde{I}$ are constructed independently, the variance of estimation of $B^p(\mathbf{s}_0, t_0)^*$ can be computed with

$$\mathrm{var}\left(B^p(\mathbf{s}_0, t_0)^*\right) = \mathrm{var}\left(M(\mathbf{s}_0, t_0)^*\right) + \sigma_I(\mathbf{s}_0, t_0)^2 + \mathrm{var}\left(\tilde{I}(\mathbf{s}_0, t_0)^*\right). \tag{11}$$

In the Gaussian process framework, $B^p(\mathbf{s}_0, t_0)^*$ and $\mathrm{var}\left(B^p(\mathbf{s}_0, t_0)^*\right)$ are referred to as the posteriori mean and variance, respectively.

The conversion of the transformed variable $B^p$ (i.e., the expected value Equation (10) and variance of Equation (11)) into bird density $B$, is possible through the use of a quantile function, $Q_B(\rho; \mathbf{s}_0, t_0)$, which returns the bird density value $b$ corresponding to a given quantile $\rho$:

$$Q_B(\rho; \mathbf{s}_0, t_0) = \inf\{b : \Pr(B(s_0, t_0) < b) \geq \rho\}. \tag{12}$$

The quantile function allows to describe $B$ because the quantile value $\rho$ is preserved through a power transform. Therefore, the quantile function of $B$ is computed with

$$Q_B\left(\rho; \mathbf{s}_0, t_0\right) = Q_{B^p}\left(\rho; \mathbf{s}_0, t_0\right)^{1/p} = \left(F_{B^p}^{-1}(\rho)\right)^{1/p}, \tag{13}$$

where $F_{B^p}(B^p)$ is the cumulative distribution function of the Gaussian variable $B^p(\mathbf{s}_0, t_0)$ characterized by its mean (Equation (10)) and variance (Equation (11)).

Consequently, we choose to characterize the estimation of the bird density $B(\mathbf{s}_0, t_0)$ by its median and its quantiles 10 and 90 (i.e., uncertainty range).

### 2.4.2. Simulation

The geostatistical simulation of the random variable $B$ consists of randomly drawing a realization $B^{(\ell)}$ among the set of all possible outcomes defined by the probability of $B$ conditional to the radar observations (see Equations (10) and (11)) e.g., [23]. This is identical to sampling the posterior probability in the Gaussian process framework.

Although kriging estimation is known to produce accurate point estimates, it leads to excessively smooth interpolation maps [28], and thus fails to reproduce the fine-scale texture of the process at hand. This causes problems for applications in which the space-time structure of the interpolation map matters, such as when a nonlinear transformation is applied to the interpolated map. This is the case in our model, as the back power transformation creates skewed distributions of bird density. Computing the total number of birds migrating is a prime example of the necessity of simulation. Indeed, integrating the bird density estimation map would greatly underestimates this number, because of its inability to reproduce peaks of bird densities. Instead, integrating multiple realizations would produce an accurate distribution of the total number of birds.

The simulation of both $M$ and $\tilde{I}$ is performed using Sequential Gaussian Simulation [29,30]. The simulation results in multiple realizations denoted as $\{M^{(\ell)}\}$ and $\{\tilde{I}^{(\ell)}\}$. The final realization of $B$ is simply computed using,

$$B^{(\ell)}\left(\mathbf{s}_0, t_0\right) = \left(t\left(\mathbf{s}_0\right) + M^{(\ell)}(\mathbf{s}_0, t_0) + \iota\left(t_0\right) + \sigma_I\left(\mathbf{s}_0, t_0\right) \tilde{I}^{(\ell)}(\mathbf{s}_0, t_0)\right)^{1/p}. \tag{14}$$

### 2.5. Validation

### 2.5.1. Cross-Validation

We tested the internal consistency of the model by cross-validation. It consists of sequentially omitting the data of a single radar, then estimating bird densities at this radar location with the model, and, finally, comparing the model-estimated value $B^p(\mathbf{s}, t)^*$ to the observed data $B^p(\mathbf{s}, t)$. The model is assessed by its ability to provide both the smallest misfit errors, i.e., $\|B^p(\mathbf{s}, t)^* - B^p(\mathbf{s}, t)\|$, and uncertainty ranges matching the magnitude of these errors. Because it is more convenient to quantify these two aspects with a normal variable, we used the transformed variable $B^p$ and quantified the model performance with the normalized error of estimation, defined as

$$\frac{B^p(\mathbf{s}, t)^* - B^p(\mathbf{s}, t)}{\sqrt{\mathrm{var}\left(B^p(\mathbf{s}, t)^*\right)}}, \tag{15}$$

where $\mathrm{var}\left(B^p(\mathbf{s}, t)^*\right)$ is the variance of the estimation as defined in Equation (11), which quantifies the uncertainty of the estimation. The numerator of Equation (15) measures the misfit of the model estimation, and the denominator normalizes this misfit according to the estimation uncertainty provided by the model. For instance, a normalized error of 1 corresponds to the estimation value being one estimated standard deviation above the measured value. Consequently, an ideal estimation should produce normalized errors of estimation that follow a standard normal distribution, because (1) the

estimation should be unbiased (mean of zero) and (2) the uncertainty provided by the model should correspond to the observed error (variance of 1).

In addition, we performed the same cross-validation procedure using a nearest-neighbor interpolation method instead, thus allowing to benchmark the proposed approach. The root-mean-square error (RMSE) and coefficient of determination $R^2$ are used to assess the performance of both the proposed model and the nearest-neighbor interpolation.

### 2.5.2. Comparison with Dedicated Bird Radars

A second validation of our modeling framework (from data acquisition by weather radars to geostatistical interpolation) requires comparing the model-predicted bird migration intensities with the measurements of two dedicated bird radars (Swiss BirdRadar Solution AG, https://swiss-birdradar.comswiss-birdradar.com) located in Herzeele, France (50°53′05.6″N 2°32′40.9″E), and Sempach, Switzerland (47°07′41.0″N 8°11′32.5″E) (see Figure 1). These radars are located 50 km and 84 km, respectively, to the closest weather radar. By comparison, the distance of the grid nodes to their closest weather radar is on average 92 km (10–90 quantiles: 35–166 km).

These bird radars continuously register echoes transiting through a conical shaped beam (17.5° nominal beam angle). The diameter of the radar beam cross-section varies from 50 m at 50 m agl to 500 m at 1500 m agl [31]. The individual echoes are aggregated over an hour to produce migration traffic rates (MTR) (bird/km/h) and an average speed of birds aloft (km/h). The bird density measured by the bird radar is computed by dividing the MTR by the mean speed [32].

Using the model presented, we estimate bird density at the exact locations of the bird radars. In order to account for the difference of time resolution, the estimations are first computed every 15 min and then averaged over one hour. We subsequently assess the quality of the estimation and uncertainty provided by the model by computing the normalized errors of estimation (Equation (9)). In addition, we also compare our approach with a simple nearest-neighbor interpolation using the RMSE and $R^2$.

## 3. Results

### *3.1. Validation*

### 3.1.1. Cross-Validation

The normalized error of estimation over all radars has a near-Gaussian distribution with a mean of 0.01 and a variance of 1.08 (Figure A7 in Appendix D). The near-zero mean of the error distribution indicates that the model provides nonbiased estimations of bird densities, where the near-one standard deviation demonstrates that the model provides appropriate uncertainty estimates. The performance of the cross-validation shows radar-specific biases (i.e., constant under- or overpredictions; Figure A8 in Appendix D and time series of each radar in Supplementary Material S2). The biases do not show any clear spatial pattern (Figure A8 in Appendix D), suggesting that these radar-specific biases probably originate from measurement errors, such as birds nonaccounted for (e.g., flying below the radar) or errors in the cleaning procedure (e.g., ground scattering).

Compared to a nearest-neighbor interpolation, the model performs better in the cross-validation with a RMSE of 17.67 bird/km$^2$ and $R^2$ of 0.59 against a RMSE of 23.17 bird/km$^2$ and $R^2$ of 0.29 for the nearest neighbor.

### 3.1.2. Comparison with Dedicated Bird Radars

The daily migration patterns estimated by the model generally coincide well with the observations derived from dedicated bird radars (Figure 4). Over the whole validation period, the normalized estimation error has a mean of 0.49 and a variance of 0.77 at Herzeele radar location, and a mean of −0.68 and a variance of 1.45 at Sempach radar location. These normalized estimation errors indicate a tendency of the model to slightly overestimate bird densities in Herzeele (i.e., mean above 1) and to underestimate them in Sempach, while providing overconfident uncertainty in Herzeele and underconfident at Sempach.

These errors translate into a RMSE of 9.2 and 19.7 bird/km$^2$ and R$^2$ of 0.87 and 0.73 for the radar in Herzeele and Sempach, respectively. Reference [32] reported relatively similar values of R$^2$ when comparing the MTR of close-by weather radar and bird radar. By comparison, the nearest-neighbor approach yields a RMSE of 9.7 and 31.7 bird/km$^2$, and a R$^2$ of 0.85 and 0.59, respectively.

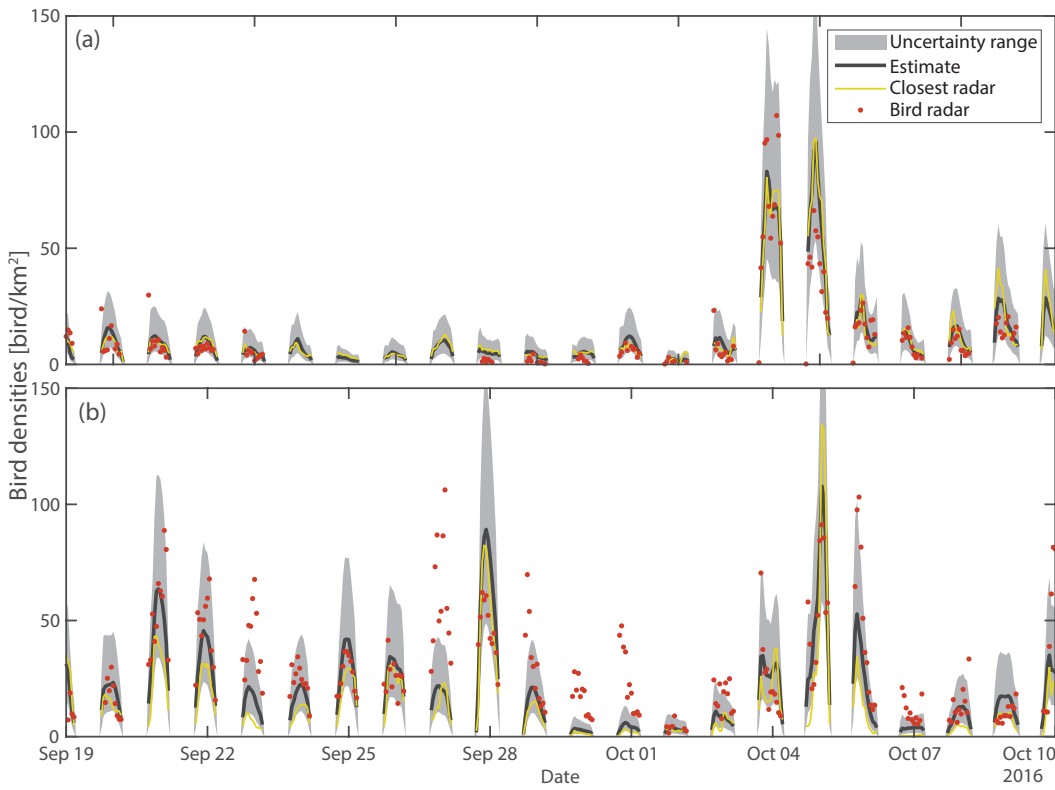

**Figure 4.** Comparison of the estimated bird densities (black line) and their uncertainty range (10–90 quantiles in gray) with the bird densities (red dots) observed using dedicated bird radars at two locations in (**a**) Herzeele, France (50°53′05.6″N 2°32′40.9″E), and (**b**) Sempach, Switzerland (47°07′41.0″N 8°11′32.5″E). Note that, because of the power transformation, model uncertainties are larger when the migration intensity is high. It is therefore critical to account for the uncertainty ranges (light gray) when comparing the interpolation results with the bird radars observations (red dots).

### 3.2. Application to Bird Migration Mapping

The main outcome of our model is to estimate bird densities at any time and location within the domain of interest. This is illustrated by the estimation of bird density time series at specific locations (e.g., Figure 4) and by the generation of bird density maps at different time steps (Figure 5).

Although the estimation represents the most likely value of bird density (i.e., mean of estimation) at each node of the grid (e.g., Figure 5), the simulation provides multiple values of bird density according to their conditional distribution and reproduces more accurately the fine-scale patterns of migration (e.g., Figure 6). As explained in Section 2.4.2, the amplitude of peak migration is more adequately illustrated in the realizations (Figure 6), compared to the smooth estimation map (Figure 5).

For each of the 100 realizations, we computed the total number of birds flying over the whole domain for each time step (Figure 7b). Within the time periods considered in this study, the peak migration occurred in the night of 4–5 October with up to 121 million (10–90 quantiles: 118–124) birds flying simultaneously. Computing this on subdomains, such as countries, highlights geographical differences in migration intensity. For instance, during the same night, France had 37 (35–39) million birds aloft (89 bird/km$^2$), Poland had 14 (13–15) million (65 bird/km$^2$), and Finland only 2 (1.9–2.2) million (30 bird/km$^2$) (Figure 7c).

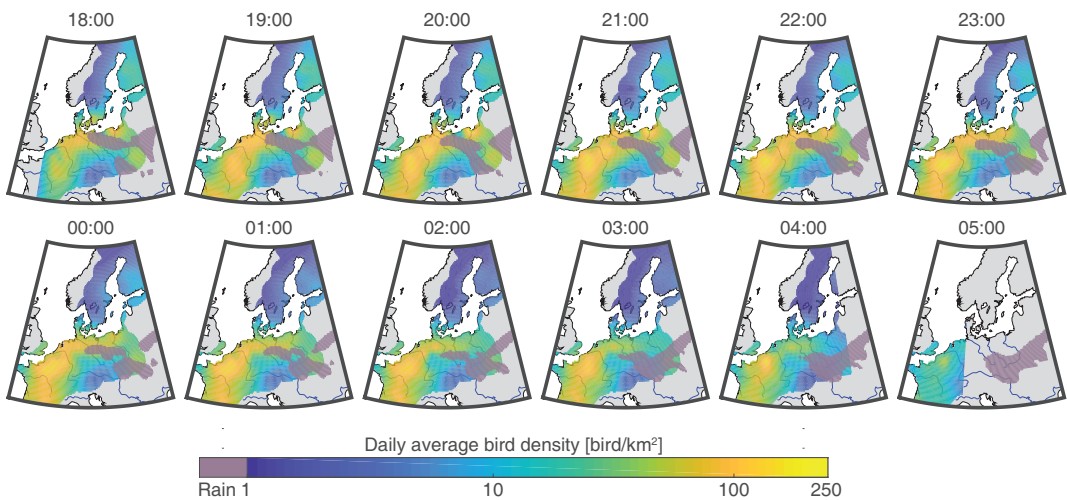

**Figure 5.** Maps of bird density estimation every hour of a single night (3–4 October). The sunrise and sunset fronts are visible at 18:00 and 05:00 with lower densities close to the fronts and no value after the front. The resemblance from hour-to-hour illustrates the high temporal continuity of the model. A rain cell above Poland blocked migration on the eastern part of the domain. In contrast, a clear pathway is visible from Northern Germany to Southwestern France.

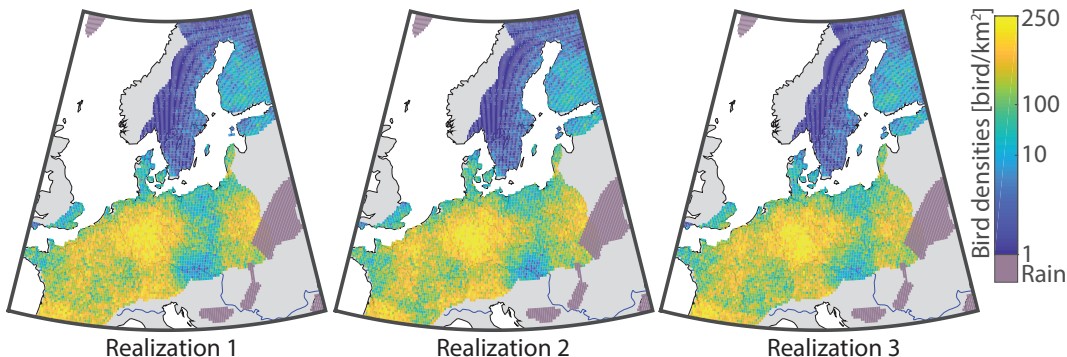

**Figure 6.** Snapshot of three different realizations showing peak migration (4 October 2016 21:00 UTC). The total number of birds in the air for these realizations was 125, 126, and 122 million, respectively. Comparing the similarities and differences of bird density patterns among the realizations illustrates the variability allowed by the stochastic model. The texture of these realizations is more coherent with the observations than the smooth estimation map in Figure 5.

The spatio-temporal dynamics of bird migration can be visualized with an animated and interactive map (available online at https://birdmigrationmap.vogelwarte.ch with a user manual

provided in Appendix E), produced with an open source script (https://github.com/Rafnuss-PostDoc/BMM-web). In the web app, users can visualize the estimation maps or a single simulation map animated in time, as well as time series of bird densities of any location on the map. In addition, it is also possible to compute the number of birds over a custom area and to download all data through a dedicated API.

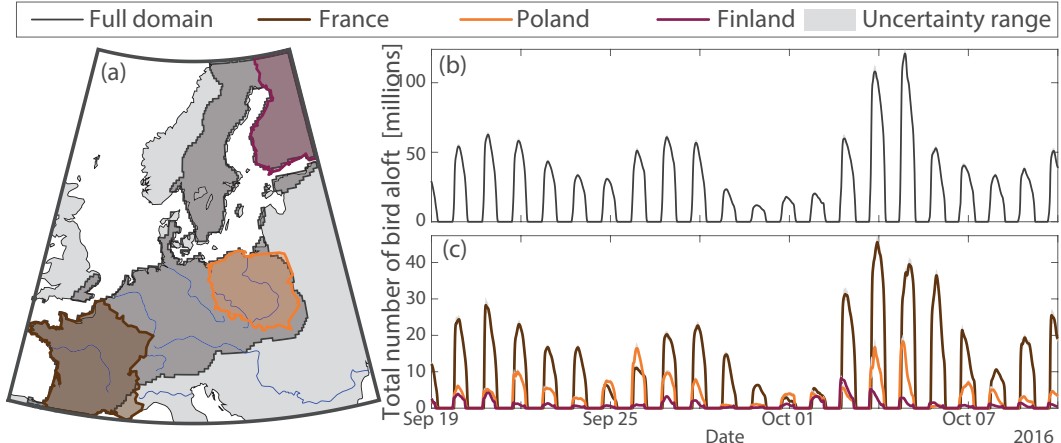

**Figure 7.** Averaged time series of birds in migration and their associated uncertainty ranges (10–90 quantiles) over (b) the whole domain, and (c) France, Poland and Finland. (a) illustrates the location and size of the three countries.

## 4. Discussion

The model developed here can estimate bird migration intensity and its uncertainty range on a high-resolution space-time grid (0.2° lat. lon. and 15 min.). The highest total number of birds flying simultaneously over the study area is estimated at 121 million, corresponding to an average density of 52 bird/km$^2$. This number illustrates the impressive magnitude of nocturnal bird migration, and resembles the values of peak migration estimated over the USA with 500 million birds and a similar average density of 51 bird/km$^2$ [19]. For more local results, interactive maps of the resulting bird density are available on a website with a dedicated interface that facilitates the visualization and export of the estimated bird densities with their associated uncertainty (https://birdmigrationmap.vogelwarte.ch, see Appendix E for a user manual).

### 4.1. Advantages and Limitations

This paper presents the first spatio-temporal interpolation of nocturnal bird densities at the continental scale that accounts for sub-daily fluctuations and provides uncertainty ranges. In contrast to the methods based on covariates, deemed more reliable for extrapolation in the future [19,33,34], our interpolation approach relies solely on the strong space-time correlation of bird migration and consequently does not require any external covariate per se (e.g., temperature, rain, or wind). Similarly, local features such as the proximity to the ocean or the presence of mountains were not explicitly accounted for in the model. Yet, the influence of these meteorological and geographical features on bird migration is largely captured by the measurements of weather radars, so that, in turn, the interpolation implicitly accounts for them. However, to take full advantage of these covariates, often available at extensive scale, the model could be adapted to consider linear relationships with covariates, using the standard frameworks of regression kriging, or possibly full co-kriging [23].

The fitted parameters of the model provide information on the broad scale bird migration (see Appendix B). In particular, the covariance function of the model describes the general spatio-temporal scale at which migration is happening (Figure A5 in Appendix B). For instance, even with the spatial trend removed, the multi-night scale of bird migration ($M$) correlates at 50% at a distance of 300 km (25%–500 km) and at 60% for one day to the next (25% at 3 days). These ranges qualitatively describe the spatio-temporal extent of broad-front migration in the midst of the autumn migration season, and highlight the importance of international cooperation for data acquisition and for the spread of warning systems during peak migration events.

As a proof-of-concept, we used three weeks of bird density data available on the ENRAM data repository (see Data Accessibility). As more data from weather radars become available, our analyses can be extended to year-round estimations of migration intensity at the continental scale where weather radar data are available with a good coverage, as is the case in Europe and North America. We also significantly preprocessed the bird density data, i.e., restricted our model to nocturnal movements, and applied a strict manual data cleaning. This is because the bird density data presently made available can be strongly contaminated with the presence of insects during the day, and birds flying at low altitude are not reliably recorded by radars because of ground clutter and the radar position in relation to its surrounding topography. Once the quality of the bird density data has improved [35], our model can be implemented in near-real-time and provide continuous information to stakeholders and the public and private sectors.

Although the model introduced here is already a valuable tool for bird migration mapping, we see several avenues for further development. For instance, in applications where the distribution of bird density over altitudes is crucial, the model can be extended to explicitly incorporate the vertical dimension. Furthermore, if fluxes of birds, i.e., migration traffic rates, are sought after, a similar geostatistical approach can be used to interpolate the flight speeds and directions that are also derived from weather radar data. If one has access to the raw radial velocity, the method developed by [36] would produce more accurate results.

### 4.2. Applications

Many applied problems rely on high-resolution estimates of bird densities and migration intensities, and the model developed here lays the groundwork for addressing these challenges. For instance, such migration maps can help identify migration hotspots, i.e., areas through which many aerial migrants move, and thus, assist in prioritizing conservation efforts. Furthermore, mitigating collision risks of birds by turning off artificial lights on tall buildings or shutting down wind energy installations requires information on when and where migration intensity peaks. The probability distribution function of our model can provide such information as it estimates when and where migration intensity exceeds a given threshold. Such information can be used in shut-down on demand protocols for wind turbine operators or trigger alarms to infrastructure managers.

**Supplementary Materials:** The following are available online at http://www.mdpi.com/2072-4292/11/19/2233/s1. The illustrations of the cleaned vertical-integrated time series of nocturnal bird densities for all radars are available in Supplementary Material S1. The illustrations of the cross-validation of all radars are available in Supplementary Material S2. The MATLAB livescripts of this article are available at https://rafnuss-postdoc.github.io/BMM. The cleaned vertical time series profile are available at doi: https://doi.org/10.5281/zenodo.3243397 [37]. and the final interpolated maps are available at doi: https://doi.org/10.5281/zenodo.3243466 [38]. The codes of the website (HTML, Js, NodeJs, Css) are available at https://github.com/rafnuss-postdoc/BMM-web.

**Author Contributions:** R.N., L.B., F.L., and B.S. conceived the study; R.N., L.B., and G.M. designed the geostatistical model; R.N. developed and implemented the computational framework; and R.N., L.B., and B.S. performed the analyses and wrote a first draft of the manuscript; G.M., F.L. and S.B. contributed substantially to the manuscript.

**Funding:** We acknowledge the financial support from the Globam project, funded by BioDIVERSA, including the Swiss National Science Foundation (31BD20_184120), Netherlands Organisation for Scientific Research (NWO E10008), and Academy of Finland (aka 326315), BelSPO BR/185/A1/GloBAM-BE.

**Acknowledgments:** This study contains modified Copernicus Climate Change Service Information 2019. Neither the European Commission nor ECMWF is responsible for any use that may be made of the Copernicus Information or Data it contains. We acknowledge the European Operational Program for Exchange of Weather Radar Information (EUMETNET/OPERA) for providing access to European radar data, facilitated through a research-only license agreement between EUMETNET/OPERA members and ENRAM (European Network for Radar surveillance of Animal Movements). Mathieu Boos (Research Agency in Applied Ecology, Naturaconsta, Wilshausen, France) kindly provided the BirdScan data for Herzeele in France.

**Conflicts of Interest:** The authors declare no conflicts of interest.

## Appendix A. Data Preprocessing

Appendix A describes the full procedure applied to manually clean the time series of bird densities. The raw dataset of bird density downloaded from the ENRAM data repository (https://github.com/enram/data-repository) has been previously published in [21]. The steps detailed below are illustrated in Figure A1 for the radar located in Zaventem, Belgium (50°54′19″N, 4°27′28″E).

1. Of the 84 radars contributing data during the study period, 11 radars are discarded because of their poor quality due to S-band radar type, poor processing, or large gaps (temporal or altitude cut). The same radars were removed in [21]. In addition, the four radars from Bulgaria and Portugal were excluded because of their geographic isolation.
2. The full vertical profile was discarded when rain was present at any altitude bin (purple rectangle in Figure A1). A dedicated MATLAB GUI was used to visualize the data and manually set bird densities to "not-a-number" in such cases.
3. Zones of high bird densities can sometimes be incorrectly eliminated in the raw data (red rectangle in Figure A1). To address this, reference [21] excluded problematic time or height ranges from the data. Here, in order to keep as much data as possible, the data was manually edited to replace erroneous data either with "not-a-number", or by cubic interpolation using the dedicated MATLAB GUI.
4. Due to ground scattering (brown rectangle in Figure A1), the lower altitude layers are sometimes contaminated by errors or excluded in the raw data. We vertically interpolated bird density by copying the first layer without error into to the lower ones. This approach is relatively conservative as bird migration intensity usually decreases with height in the absence of obstacles, and more so in autumn [39].
5. The vertical profiles were vertically integrated from the radar ground level (black line in Figure A1c) and up to 5000 m asl.
6. The data recorded during daytime are excluded. Daytime is defined for each radar by the civil dawn and dusk (sun 6° below horizon).
7. Finally, the data of 10 radars with high temporal resolution (5–10 min) was downsampled to 15 min to preserve a balanced representation of each radar.

The resulting cleaned time series of nocturnal bird densities are displayed in Figure A1d. We provide the same figure than Figure A1 for all radar data in Supplementary Material S1.

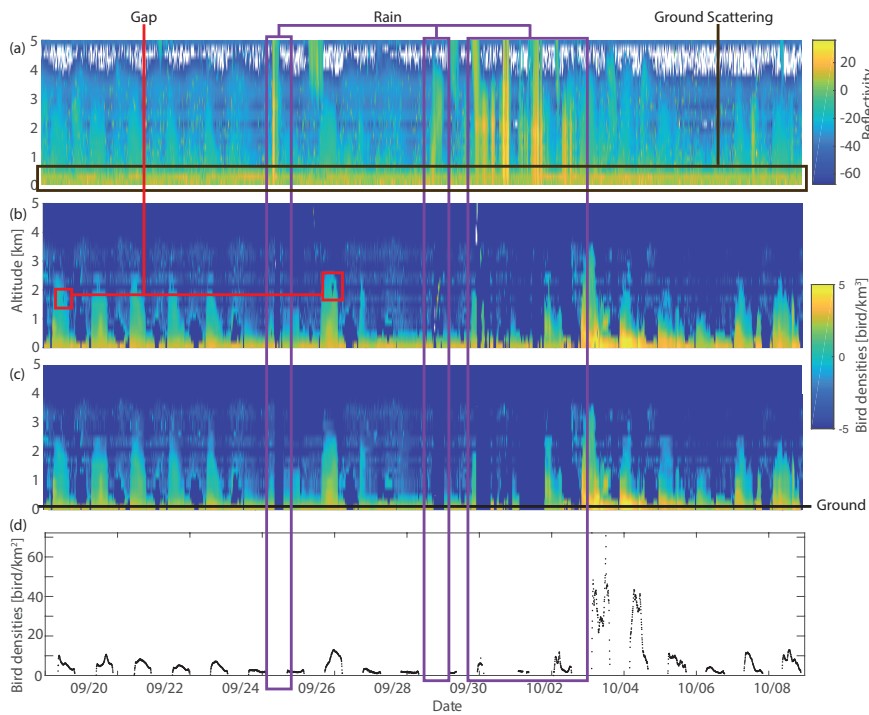

**Figure A1.** Illustration of the cleaning procedure for the data of the Zaventem radar in Belgium (50°54′19″N, 4°27′28″E). (**a**) Raw reflectivity. (**b**) Raw data of vertical bird density profiles. (**c**) Manually cleaned vertical bird profiles. (**d**) Final bird densities (integrated over all altitudes).

## Appendix B. Model Parametrisation

Appendix B presents the method of model parametrisation and discusses the meaning of model parameters in terms of bird migration. Table A1 summarizes the calibrated parameters.

**Table A1.** Calibrated parameters.

| | |
|---|---|
| **Power transformation** | $p = 0.133$ |
| **Spatial trend** | $w_0 = 2.566$ , $w_{\text{lat}} = -0.024$ |
| **Covariance of** $M$ | $C_0 = 0.006, C_g = 0.032, r_t = 1.24, r_s = 500, \alpha = 0.98, \gamma = 0.71, \beta = 0.95$ |
| **Curve** | $a = [0.04, -0.10, 0.07, 0.27, -1.29, -0.59, 2.86, 0.44, -1.92]$ |
| **Curve variance** | $b = [0.00, 0.00, 0.02, 0.04, -0.17, -0.17, 0.62, 0.26, -0.93, -0.12, 0.49]$ |
| **Covariance of** $I$ | $C_0 = 0.009, C_g = 0.91, r_t = 0.07, r_s = 190, \alpha = 1, \gamma = 0.4, \beta = 1$ |

*Appendix B.1. Power Transform p*

The value of power transformation $p$ is inferred by minimizing the Kolmogorov–Smirnov criterion of the p-transformed observations $B(\mathbf{s}, t)^p$. The Kolmogorov–Smirnov test (Massey, 1951) assesses the hypothesis that the observations $B(\mathbf{s}, t)^p$ are normally distributed. The optimal power transformation parameter was found for $p = 1/7.5$ and the resulting $B^p$ histogram is displayed in Figure A2 together with the initial data $B$.

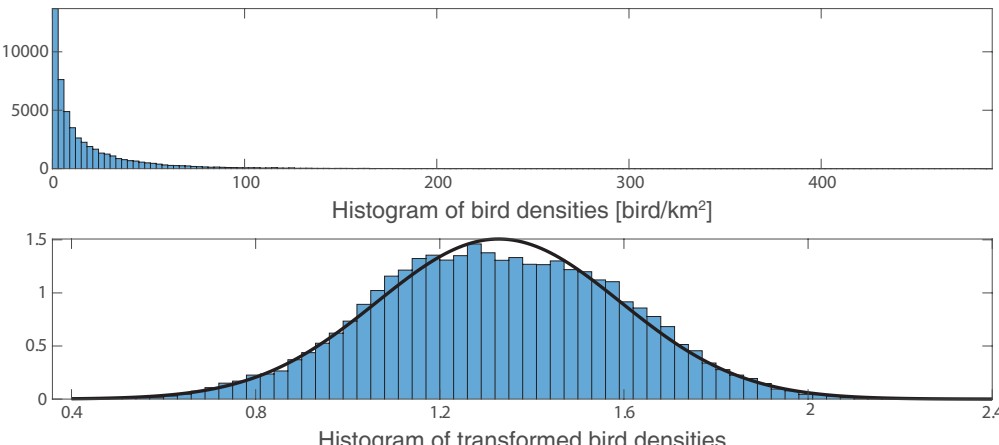

**Figure A2.** Histogram of the raw bird densities data $B$ (**top**) and the power transformed bird densities $B^p$ for the calibrated parameter $p = 1/7.5$ (**bottom**).

The fitted distribution shows that the distribution of bird densities is highly skewed: the lowest 10% are below 1 bird/km$^2$, the upper 10% are above 50 bird/km$^2$, and the maximum density reaches 500 bird/km$^2$. Consequently, the central value (mean of 19 bird/km$^2$ or median of 8 bird/km$^2$) and variance (753 bird$^2$/km$^4$) do not adequately characterize the distribution. A power transformation on such skewed data creates significant nonlinear effects in the back-transformation. For instance, the symmetric uncertainty of an estimated value in the transformed space (quantified by the variance of estimation) will become highly skewed in the original space. Consequently, the uncertainty of the estimation of bird densities is highly dependent on the value of the power transform: low densities estimations have a smaller uncertainty than high densities.

Such effects also have consequences from an ecological and conservation point of view. Indeed, efficient protection of birds along the migration route (from artificial light or wind turbines) requires particular attention to the peak densities, during which the majority of birds are moving in a few nights. These peaks can only be accurately reproduced by accounting for the high tail of the distribution. This is done here by using a full distribution to quantify the uncertainty of the estimation.

*Appendix B.2. Spatial Trend μ*

The parameters of the spatial trend ($w_{\text{lat}}$ and $w_0$) are calibrated on the nightly average of each radar with an ordinary least square method. The resulting planar trend is shown in Figure A3a together with the average transformed bird densities of each radar. The trend displays a strong north–south gradient, which can be explained by the higher migration activity in southern Europe during the study period. A 2-dimensional planar trend was initially tested in order to accommodate the northeast–southwest flyway. However, this more complex model did not significantly improve the fit with data, and was therefore discarded. The detrended values illustrated in Figure A3b are more stationary, with the exception of Finland and Sweden. Reference [21] also noted this difference between these countries, but excluded the fact that this difference is due to errors in the data since the southern Swedish radar shows consistent amounts of migratory movements with a neighboring German site. Figure A3b highlights the central European continental flyway as illustrated by the arrow.

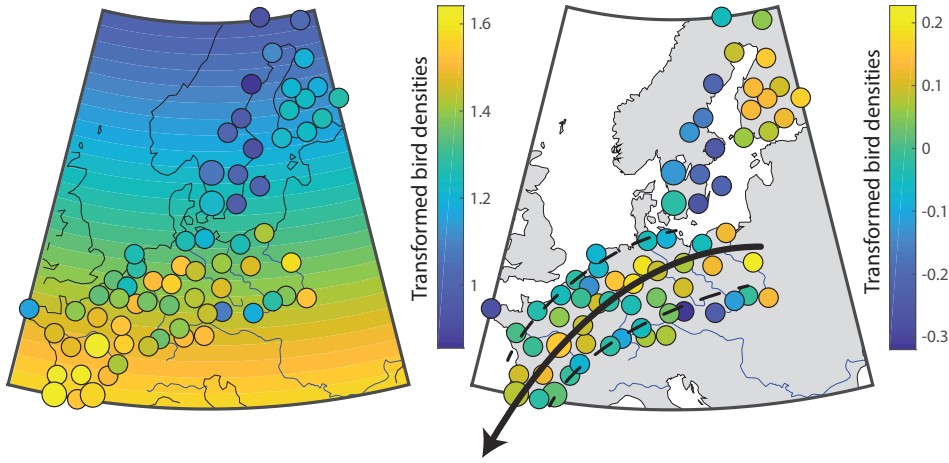

**Figure A3.** Fitted trend with averaged observations at radar location (**left**) and detrended data (**right**).

*Appendix B.3. Curve Trend ι and Variance σ_I*

Figure A4 displays the intra-night scale component $B^p - \mu - M$ of the weather radar data together with the fitted polynomial curve trend $\iota$ (Equation (6)) and polynomial variance function $\sigma_I$ (Equation (8)). The curve reveals that migration is mainly concentrated between 10% and 90% of the night-time with slightly larger densities of birds in the first half of the night. The larger variance at the beginning and end of the night is partly due to the power-transformation of the raw data.

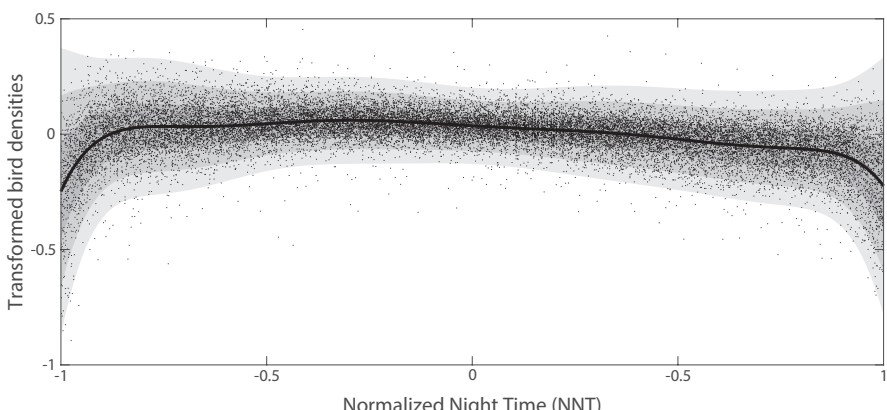

**Figure A4.** Intra-night scale component of the weather radar data (black dots) and fitted curve trend (black line). The shaded gray areas each denote 1-, 2-, and 3-$\sigma_I$ fitted.

*Appendix B.4. Covariance Functions of M and I*

The parameters of the space-time covariance functions ($C_0, C_G, r_t, r_s, \alpha, \gamma$, and $\beta$) of $M$ (Equation (4)) and $I$ (Equation (9)) are inferred by minimizing the misfit between the covariance function and the empirical covariances of the weather radar data. The empirical covariances are derived for several lag-distances and lag-times on an irregular grid.

The calibrated covariance functions provide some information about the degree of spatial and temporal correlation of the bird migration process. Indeed, the absolute value of bird density covariance should also include the effect of trend and power transformation. The fitted spatial covariance at the multi-night scale $M$ (Figure A5a) is fitted with a large spatial nugget (16%), suggesting a significant variability of bird density uncorrelated in space. This can be explained by either local features of the migration process or radar measurement errors. It is important to recall that since the weather radars are relatively well-spread (Figure 1b), the spatial covariance of both $M$ and $I$ is poorly constrained for small lag-distances (approximately less than 100 km), and consequently the uncertainty of the nugget value is large. The temporal covariance of $M$ has an asymptotic behavior with 20% covariance after

6 days (Figure A5b) because no temporal trend was used. Note that as the covariance of $M$ is evaluated only on a discrete 1-day lag-time, the shape of the covariance between 0 and 1 is artificially created to fit the Gneiting function. $M$ and $\mu$ account for most of the spatial covariance of bird density, but some spatial correlation is still present at the intra-night scale as suggested by Figure 1c. The temporal correlation of $\tilde{I}$ is high for short lags (67% at 0.04 days, or 1 hr), indicating consistent measurements from each weather radar.

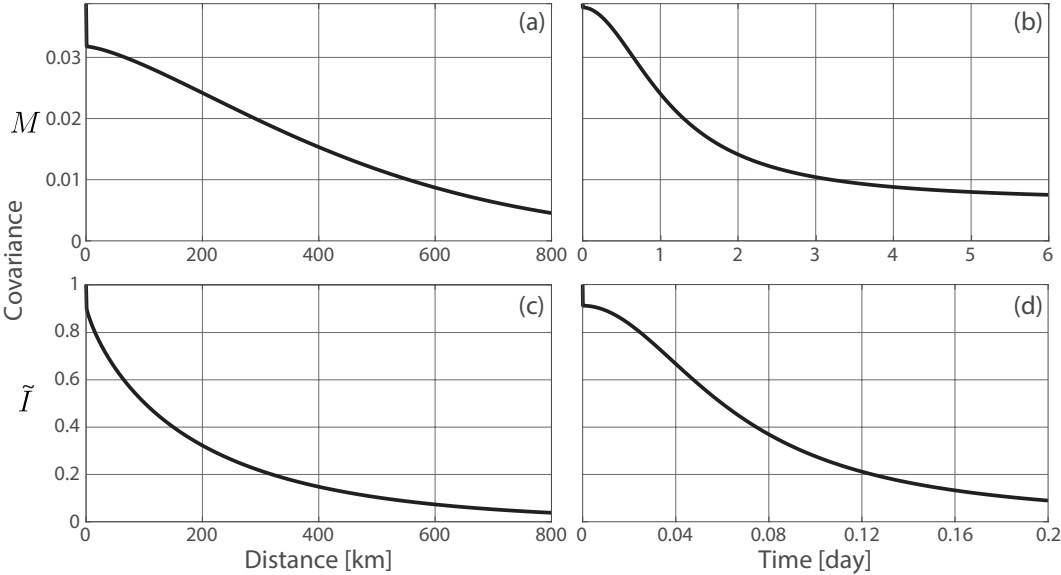

**Figure A5.** Illustration of the calibrated covariance function of the multi-night scale $M$ (**a**,**b**) and the intra-night scale $\tilde{I}$ (**c**,**d**).

## Appendix C. Kriging

Appendix C provides detailed explanations for the kriging estimation. Note that this procedure is identical to Gaussian regression in the field of machine learning, where the kriging estimation is equivalent to the mean of the posterior distribution (i.e., conditional to the known location).

Kriging provides an estimated value of a random variable $X$ (either $M$ or $I$) at the target point $(\mathbf{s}_0, t_0)$ based on a linear combination of known data points $\{X(\mathbf{s}_\alpha, t_\alpha)\}_{\alpha=1,\ldots,n_0}$ with

$$X(\mathbf{s}_0, t_0)^* = \sum_{\alpha=1}^{\alpha=n_0} \lambda_\alpha X(\mathbf{s}_\alpha, t_\alpha). \tag{A1}$$

The kriging weights $\mathbf{\Lambda} = [\lambda_1, \cdots, \lambda_{n_0}]^T$ are determined by minimizing the variance of the estimation under unbiased conditions which leads to the following linear system, commonly referred to as the kriging system,

$$\mathbf{C}_{\alpha,\alpha}\mathbf{\Lambda} = \mathbf{C}_{\alpha,0}, \tag{A2}$$

where $\mathbf{C}_{\alpha,\alpha}$ is the cross-covariance matrix of the observations and $\mathbf{C}_{\alpha,0}$ is the covariance matrix between the observations and the target point. These covariances are computed using the fitted covariance function of Equation (4) or Equation (9). The kriging weights can be solved using $\mathbf{\Lambda} = \mathbf{C}_{\alpha,\alpha}^{-1}\mathbf{C}_{\alpha,0}$, and are subsequently used in Equation (A1) to compute the kriging estimate. Figure A6 illustrates the value of the kriging weights for the kriging estimation of the multi-night scale at an unknown location (red dot).

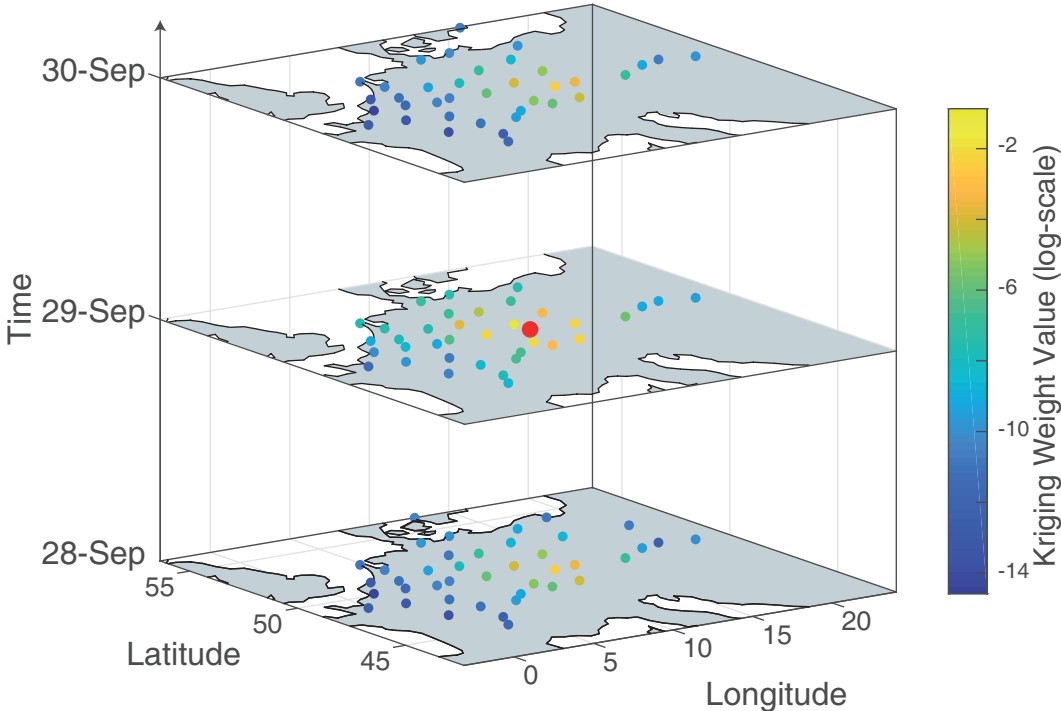

**Figure A6.** Illustration of the kriging weight values computed for the kriging estimation of *M* at an unknown location (red dot). Here we illustrate only the neighbors within +/−1 day and a spatial extent of 600 km.

In addition to the estimated value $X(\mathbf{s}_0, t_0)^*$, kriging also provides a measure of uncertainty with the variance of estimation,

$$\text{var}\left(X\left(\mathbf{s}_0, t_0\right)^*\right) = C_X\left(\mathbf{0}, 0\right) - \boldsymbol{\Lambda}^{\top} C_{\alpha,0} \tag{A3}$$

## Appendix D. Cross-Validation

Appendix D provides additional details on the cross-validation described in Section 2.5.1. Figure A7 displays the histogram of the normalized error of estimation (Equation (15)) when all data across all radars are considered. The mean of the distribution is close to zero which indicates that the estimation is unbiased (i.e., on average, the estimation neither underestimates (mean below 1) nor overestimates (mean above 1) bird density). Its variance is slightly above 1, which indicates a small tendency to underestimate the uncertainty range (i.e., on average, the kriging variance is too small). Overall, the cross-validation indicates a good performance of the model.

In order to test radar-specific performance, the normalized error of estimation is computed for each radar. As displayed in Figure A8, their respective mean and variance do not reveal any clear spatial pattern, thus suggesting no spatial biases in the estimation. However, the large absolute value of the mean of the normalized error of estimation (color scale) indicates a strong variability in the model performance for each radar. The model underestimates or overestimates bird density at some radar locations with a mean of 1.5 times the variance of estimation. The time series comparing estimations of the cross-validation with the observed data for each radar are provided in Supplementary Material S2.

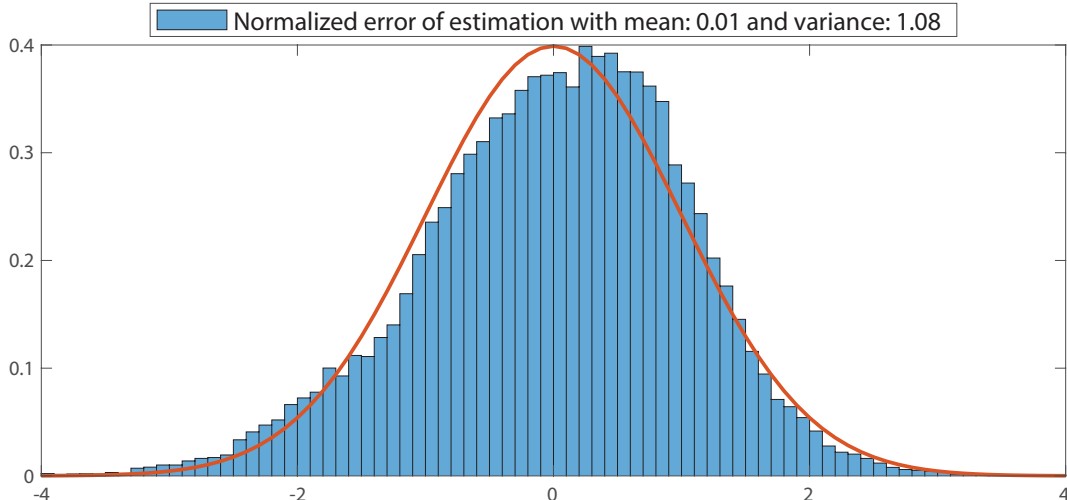

**Figure A7.** Histogram of the normalized error of estimation (Equation (15)) over all radars. The red curve is the standard normal distribution which should be matched by the histogram in case of ideal estimation.

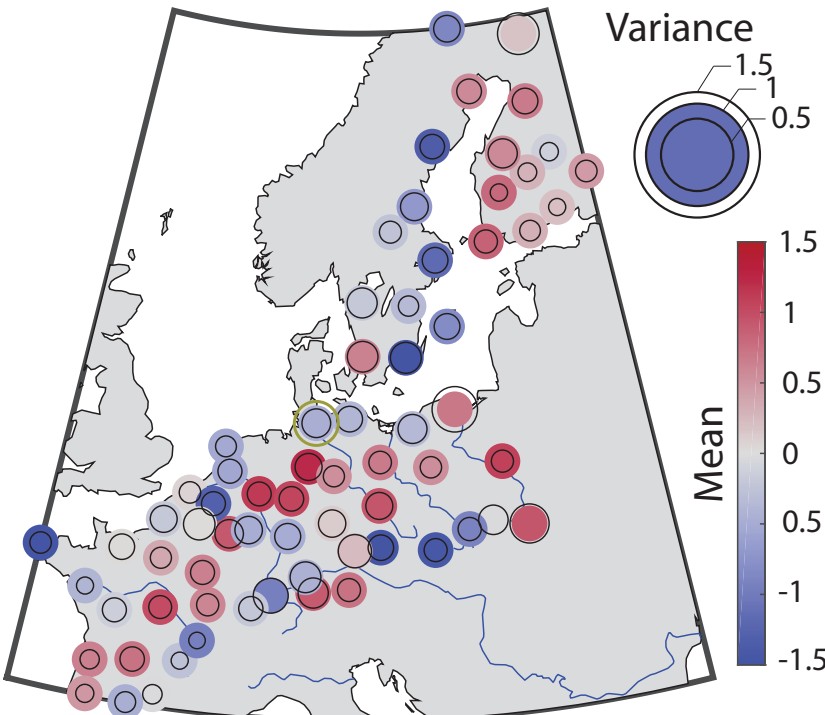

**Figure A8.** Illustration of the performance of the cross-validation with the mean and variance of the normalized error of estimation (Equation (15)) at each radar location. Negative or positive mean values respectively indicate an under- or overestimation of the model estimation. The reproduction of the variance is illustrated by a black circle: a perfect variance would match the color circle while a smaller circle indicates an underconfidence (uncertainty range too large).

**Appendix E. Manual for Website Interface**

Appendix E describes the web interface developed for the visualization and querying of the interpolated data. The website is available on https://birdmigrationmap.vogelwarte.ch and the code on https://github.com/Rafnuss-PostDoc/BMM-web. The web interface is displayed in Figure A9 with annotations and further details below.

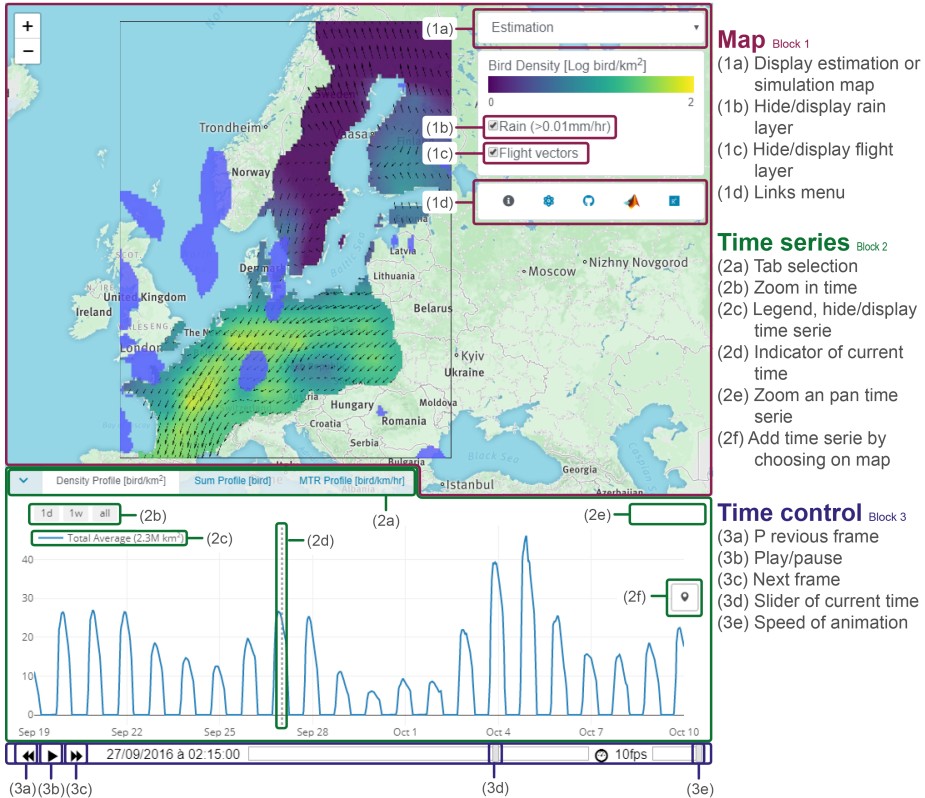

**Figure A9.** Illustration of the website interface with annotations for each interactive component.

*Appendix E.1. Block 1: Interactive Map*

The main block of the website is a map with interactive visualization tools (e.g., zoom and pan). On top of this map, three layers can be displayed:

- The first layer illustrates bird densities in a log-color scale. This layer can display either the estimation map or a single simulation map. Users can choose using the drop-down menu (1a).
- The second layer displays the rain in light-blue. The layer can be hidden/displayed using the checkbox (1b).
- The third layer corresponds to bird flight speed and direction, visualized by black arrows. The checkbox (1c) allows users to display/hide this layer. Finally, the menu (1d) provides a link to (1) documentation, (2) model description, (3) Github repository, (4) MATLAB livescript, and (5) Researchgate page.

*Appendix E.2. Block 2: Time Series*

The second block (hidden by default on the website) shows three time series, each in a different tab (2a):

- Densities profile shows the bird densities [bird/km$^2$] at a specific location.
- Sum profile shows the total number of birds [bird] over an area.
- MTR profile shows the mean traffic rate (MTR) [bird/km/h] perpendicular to a transect.

A dotted vertical line (2d) appears on each time series to show the current time frame displayed on the map (Block 1). Basic interactive tools for the visualization of the time series include zooming on a specific time period (day, week or all periods) (2b) and general zoom and auto-scale functions (2e). Each time serie can be displayed or hidden by clicking on its legend (2c). The main feature of this block is the ability to visualize bird densities at any location chosen on the map. For the densities profile tab, the button with a marker icon (2f) allows users to plot a marker on the map, and displays the bird densities profile with uncertainty (quantile 10 and 90) on the time series corresponding to this

location. Users can plot several markers to compare the different locations (Figure A10). Similarly, for the sum profile, the button with a polygon icon (2f) allows users to draw a polygon and returns the time series of the total number of birds flying over this area (Figure A11). For the MTR tab, the flux of birds is computed on a segment (line of two points) by multiplying the bird densities with the local flight speed perpendicular to that segment.

*Appendix E.3. Block 3: Time Control*

The third block shows the time progression of the animated map with a draggable slider (3d). Users can control the time with the buttons play/pause (3b), previous (3a) and next frame (3c). The speed of animation can be changed with a slider (3e).

*Appendix E.4. API*

An API based on mangodb and NodeJS allows users to download any time serie described in Block 2. Instructions can be found on https://github.com/Rafnuss-PostDoc/BMM-web.

*Appendix E.5. Examples*

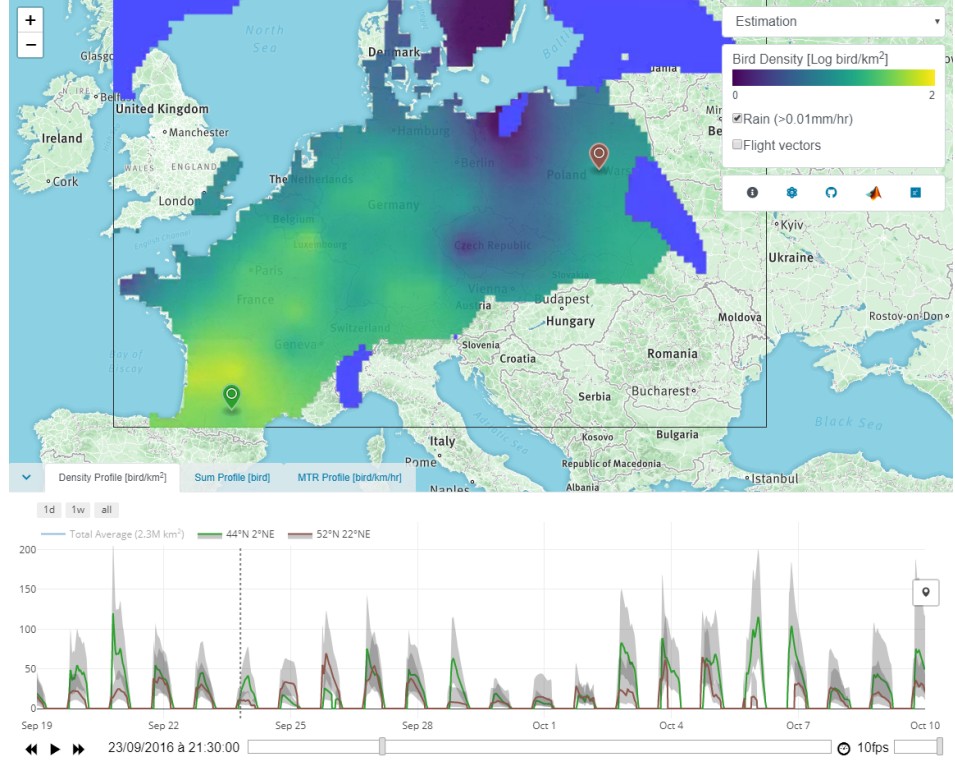

**Figure A10.** A print-screen of the web interface developed to visualize the dataset. The map shows the estimated bird densities for the 23 September 2016 at 21:30 with the rain mask appearing in light blue on top. The domain extent is illustrated by a black box. The time series in the bottom show the bird densities with quantile 10 and 90 at the two locations symbolized by the markers with corresponding color on the map. The button with a marker symbol on the right side of the time series allows users to query any location on the map and to display the corresponding time series.

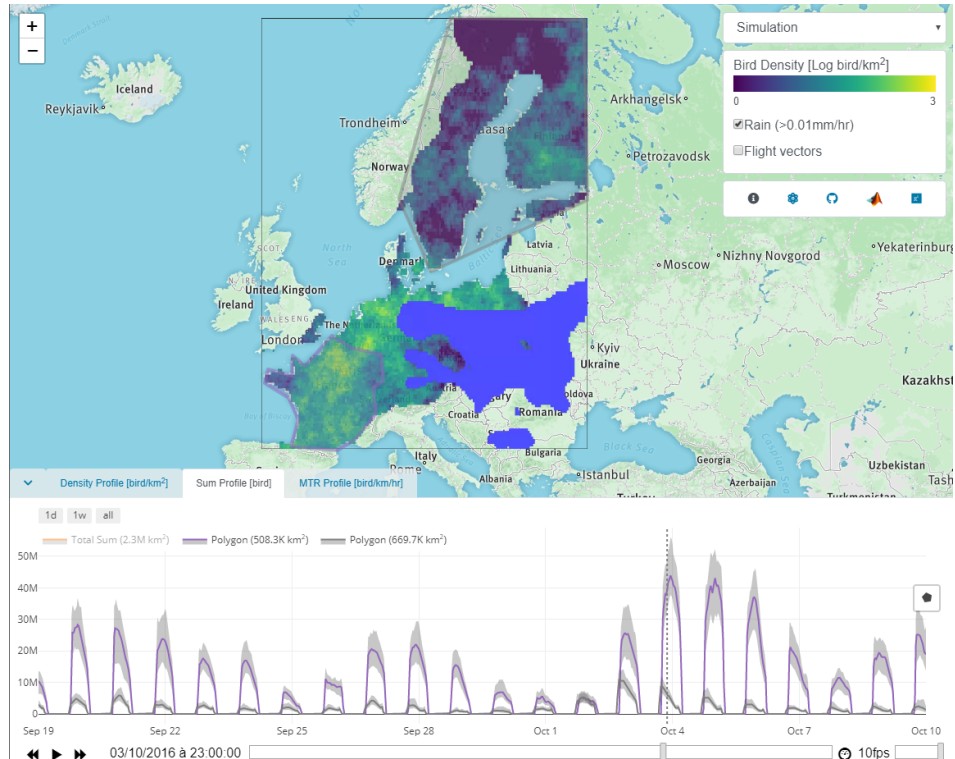

**Figure A11.** Print-screen of the web interface with the simulation map for the 3 October 2016 at 23:00. The bottom pane shows the time series of the total number of birds within the color-coded polygon drawn on the map. The button with the polygon symbol on the right side of the time series allows to query the total number of birds flying within any polygon drawn on the map.

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
