# Peer review of "A Geostatistical Approach to Estimate High Resolution Nocturnal Bird Migration Densities from a Weather Radar Network"

_remotesensing, doi:10.3390/rs11192233_

Round 1
Reviewer 1 Report
This is a promising study about interpolation of bird migration densities using weather radar data. The research has potential to be impactful. Finding appropriate methodologies to interpolate measurements in the immediate vicinity of radar stations to a fine spatial grid can enhance the usefulness of weather radar data. Kriging (i.e. Gaussian Process regression) is an appropriate methodology for doing so — it provides a principled way to interpolate and produce uncertainty estimates. The basic methodology is standard in spatial statistics. The present study does a good job approaching the problem from the applied perspective from the lens of bird migration modeling. I believe many of the practical modeling tricks the authors implement — the power transformation of the response variable, the additive modeling of the mean process in terms of spatial trend and nightly “curve”, as well as the additive decomposition of the random process in terms of nightly “amplitude” and residual — seem appropriate and can inform future research about modeling bird migration in Europe and elsewhere, especially North America where broad-scale radar data is more readily available. The paper is accompanied by an interactive website, which demonstrates that this work will not remain in a lab and has the possibility to have broader impact on potential users of this data.
The primary weaknesses of the study are in the presentation and details of the geostatistical approach. Several methodological issues need to be clarified and possibly addressed. The manuscript also needs some editing for language, grammar, and readability.
Geo-Statistical Model Development
Kriging is a classical approach, originally from the field of geo-statistics but now essentially coincides with the study of Gaussian processes across a much broader range of areas, including spatial statistics, stochastic processes, and machine learning. The presentation in this paper follows what seems to be a classical geo-statistics view of things. This is not a problem per se, but I think the paper could benefit from a slightly broader perspective. Several aspects of the presentation seem imprecise and/or mischaracterize statistical issues (see below). I think the broader perspective could help avoid this. See, for example, the books by Cressie, Wikle and Cressie, or Rasmussen and Williams.
On Line 77, the authors should state they are focusing in particular on a Gaussian process — otherwise it is not fully characterized by its mean and covariance function. The phrase “covariance function” should be used instead of “covariance matrix”. And it should be added that a Gaussian process is fully characterized by its mean function and covariance function, not just the covariance function.
I don’t agree with the emphasis on non-stationarity as a major methodological motivation made in lines 78–83 and repeated later in the paper. Non-stationarity is indeed recognized as a more challenging case for spatiotemporal modeling, but only when the covariance function is non-stationary. It becomes clear later that the authors only consider a non-stationary mean function. This is not substantially more difficult than the zero-mean or constant-mean case. In fact, the authors propose what I believe is a non-standard approach to estimating the mean function, when in fact it could be handled using a standard approach. The authors (in the appendix) describe a two-step approach where first the parameters of the trend and curve are estimated using a local search technique, and then residuals are fit using a Gaussian process (or kriging). In fact, both the trend and curve are modeled as linear models in spatio-temporal basis functions (latitude, and the first 6 powers of NNT(s,t)). This problem fits into the standard “universal kriging” or “regression kriging”. The standard solution approach boils down to solving an ordinary least squares problem followed by simple kriging (with a zero mean function) on the residuals. It is similar to what the authors propose, except it uses OLS instead of a generic optimization, and it has well understood mathematical properties. I strongly suggest the authors describe the problem in these more standard terms and adopt standard methodology or justify more clearly why they choose a non-standard terminology and/or methods.
The development of the random processes A(s,t) and R(s,t) could also use some refining. I suggest noting following Eq. (1) that A and R are assumed to be independent and zero-mean. The “curve” is sometimes denoted \gamma and sometimes c (line 86, 106, and Eq. B9, at least). Since A and R are independent and additive, their role is to determine the covariance function and they could clearly be combined into one process. The authors should provide more justification why they are kept separate. I believe it is essentially a modeling choice to encode some prior structure of temporal variation at different scales into the model (though I believe this could also be done directly in the covariance function). The overall model development was hard to read. Sometimes too few details were given in the text, and sometimes too many, causing a lot of flipping back and forth from the main text to the supporting information.
Does the definition of NNT need an equation (Eq. 4) in the main text? I believe its easier to understand without the equation and the space would be better used by including more information about the statistical model.
The need to estimate the amplitude values for individual radars and nights (lines 404-405) is puzzling. Since the combined process A(s,t) + R(s,t) is Gaussian with a known covariance function (the sum of the two individual covariance functions), I believe all reasoning about this random portion of the process can be handled through kriging. Please justify why it is necessary to estimate random values as parameters.
On line 346 and in Eq. (B5) you imply that the prediction at point (s0, t0) will depend only on the n0 nearest neighbors. In general I believe it should depend on all observed points. Please clarify why it only depends on some of the points. Is this a computational simplification?
In Eq. (B8), what is C_{ff,0}? Does the superscript t in $\Lambda^t$ mean the transpose of the matrix? (I am more accustomed to a capital “T” or \top.)
I couldn’t figure out the standardization — in particular, the meaning of $\sigma_A$ and $\sigma_R(s, t)$ in Eqs. (B1) and (B2). You say that $\sigma_A$ is the “empirical standard deviation of A”. Since A is unobserved and random, it’s unclear to me what this means. Perhaps you mean the empirical standard deviation of the conditional mean of A at the grid locations conditioned on the observed values? The interpretation of $\sigma_R(s, t)$ was even less clear to me — Eq. (B3) makes it seem like this is an additional non-random quantity you are estimating. If so, please clarify how it is estimated.
In Eqs. (B8–B10) and the surrounding text, it should be clarified that the expectation and variance are conditional on the observed values of the random process. These are sometimes described as the posterior mean and variance.
Most or all of the text in Lines 365–376 is extraneous. For any scalar random variable Z and positive value p, Eq. (16) is true by the identity Pr[Z <= z] = Pr[Z^p <= z^p]. (If it were needed, Eq. (B14) is a very standard change-of-variables formula that can be found in many textbooks.)
I’m surprised by the phrase “fminsearch function of MATLAB which uses a simplex algorithm”. First, I don’t believe fminsearch uses a simplex algorithm. Second, I doubt a simplex algorithm is needed or appropriate, since that applies to optimizing a linear function over a (linearly) constrained set. I believe your objective is a squared-error and hence non-linear (quadratic), and there are no constraints. Your problem can probably be solved by OLS.
In Line 382, do you mean “minimizing” instead of “maximizing” for the Kolmogorov-Smirnov criterion? Usually the KS statistics is smaller when the distributions are better matched.
The discussion in lines 141—148 and meaning of “geostatistical simulation” are unclear to me, as is the meaning of “simulation” in Line 205. After some puzzling, I think that a “simulation” is one sample from the posterior Gaussian process (i.e., the Gaussian process conditioned on the observed values), which is expected to be less smooth than the posterior mean (i.e. the kriging estimate). I suggest you incorporate some of this language into your description. It will help it reach a much broader audience. If this is the correct interpretation, then the samples would not be “equiprobable” (i.e. they would not have the same density under the posterior distribution).
In Lines 242–244, you describe “kriging with covariates” as “co-kriging”. I believe this is incorrect. Co-kriging refers to kriging with multiple correlated response variables. I believe the appropriate term for “kriging with covariates” is “universal kriging” or “regression kriging”. As mentioned earlier, I think you are already doing this by modeling the mean function as a linear function of spatio-temporal basis functions. I disagree, or don’t understand, the statement on Lines 244-246 about adding covariates only being helpful if they are more correlated with bird densities than the spatio-temporal correlation.
I would use more caution about interpreting the length scales of the covariance function (Lines 247–254). These are contingent on your choice of parametric form for the covariance functions, which may or may not be “correct”, and there could be a wide range of different covariance functions with different properties that provide similar fits to the data. It may be easier to make these statements using empirical covariograms instead of fitted ones.
Bird Migration
The study is strong from a bird migration standpoint. I have only a few comments and questions.
The main text says profiles are manually cleaned. Appendix A Line 314 and Figure A1 also refer to an “automatic” cleaning procedure. Please clarify the nature of this automatic procedure, and whether the cleaning is manual, or partly automatic.
I didn’t think TOO hard about the meaning of the normalized errors in the cross-validation but believe these provide useful information about the tendency to over or under estimate bird densities as well as the general uncertainty calibration. But don’t the absolute errors also convey additional useful information? Specifically, how accurately can one held out radar be estimated from the others?
It’s nice to have a comparison to the dedicated bird radars. Please report the distance from the bird radars to the closest weather radar — this would help put the performance in context. Also, as baseline, it would be very useful to report how accurately the bird radars can be predicted by only using the data from the closest weather radar. It will help justify the extent to which the geostatistical model improves point estimation relative to a simple approach such as nearest-neighbor interpolation. (Note: I believe the smooth estimates and uncertainty estimates are valuable even if point estimates are not better than a simple approach.)
It’s exciting to be able to produce within-night maps like the ones in Figure 5, but there is relatively little change from hour-to-hour, which makes me question the usefulness of this figure.
Minor Points
— Fig. 3: Perhaps provide more text explanation. The dashed line is not explained in the legend. It may also help the reader understand this plot better if you clarify that the curve and amplitude can take negative values. It’s tempting to think of the “stacked lines” that you labeled on Sep 27 to illustrated the additive decomposition as always being positive.
— Line 94 “non-stationary” —> “non-stationarity”
— Line 128. Suggested rewording: “estimation of bird density Z(s0, t0)^* at any unsampled location (s0, t0) is performed…”
— Line 138. “Uncertainty range is defined as the quantiles 10 and 90”. Reword. “quantiles 10 and 90” is not obviously a “range”
— Line 150. Add space after 0.2 degrees
— Line 158. “value and Z(s, t)^*” —> “value Z(s,t)^* and”
— Line 176. The mean is reported as 0.017 in the main text and –0.017 in the appendix.
— Line 180. Use semicolon to separate parentheticals instead of two sets of parentheses.
— Line 180. “The biases are not spatially correlated”. This is too strong a statement to make from the visual evidence in the figure.
— Lines 182–184, discussion of possible sources of biases. Perhaps move to discussion.
— Lines 192–193. “normalized estimation error has mean 0.6 and variance of 1.3”. The normalized errors are challenging to interpret. I suggest including a primer explaining why normalized errors with variance greater than 1 actually corresponds to underestimating uncertainty. I could figure this out only after some puzzling. The method of estimating the denominator in Eq. (5) is still unclear to me so I still cannot fully interpret these values.
— Lines 198-199. What is the overall mean and variance of bird densities? It would help to understand that RMSE of 20 birds/km^2 is “good performance” to have this context.
— Figure 4 caption: missing comma between “10-90 quantiles” and “uncertainty range”.
— Figure 5 caption. There is some editing error here. Each point is made twice using different language.
— Line 205: “a simulation generate”. correct wording
— Lines 214-215: the quantile ranges “[39–51]” and “[15–25]” are formatted very similarly to citations. Consider using regular parentheses to avoid this confusion.
— Nearly the same point is made in Lines 217–223 and then just a few lines later in Lines 230–233.
— Line 416 “improved” —> “improve”
Reviewer 2 Report
This paper presents a new detailed method for interpolating bird migration across space and time using data from a monitoring network of Doppler radars. This is a timely and well-presented paper. Although Doppler weather radars have been used in the United States to study bird migration for decades, the integration of radar data from different countries to track continental bird migration across Europe is an exciting new advance, and to my knowledge this is the most detailed approach for estimating bird migration intensities across space and time across any continental extent. My background is in biology, and the many potential applications of this tool are exciting. In my review of the paper, I was less able to verify the correctness of the presented equations and statistics, but the work was clearly undertaken with great attention to detail. I recommend publication of this manuscript. Here are additional minor thoughts:
Line 51-52: Please define a “bird radar,” as most readers will not be familiar with this term and how “bird radars” compare with weather radars.
Figure 2: I found the link between the colors of the outer circles in (a) and the colors of the points in (b,c,d) to be somewhat confusing. One options could be to pick three very different colors and repeat them for each panel (b,c,d).
Line 86: You refer to ‘c’ as the curve describing the nightly trend, but equation (1) includes no ‘c’.
Line 87: In “curve describing the nightly trend,” I recommend using a word other than “trend” to describe the variation through the night, because that could cause confusion with the “continental trend.”
Lines 206-209: I found it difficult to understand this sentence. Why couldn’t the estimation reproduce the number of birds during peak migration?
Figure A1: It is unclear what the unlabeled green box on panel (a) represents. Presumably it is ground clutter.
Lines 433 and figure C4: You state that the residuals signal is “spatially uncorrelated,” but on Figure C4 (a), it is clear that there is spatial variation. For example, southerly radars are generally red and northerly radars are generally blue.
Reviewer 3 Report
You have developed and tested a sophisticated geostatistical (i.e., kriging) model to interpolate observed bird migration intensity monitored by a network of weather radars across Europe. The model decomposes spatial and temporal non-stationarity of bird density aloft into four independent components. The model is novel and the four components are well-described and make biological sense from a theoretical (i.e. first-principles) perspective.
Two methods of validation of the model were used 1) cross-validation of the weather radar dataset collected over a 3-week time period to assess internal validity of the model and 2) comparison of model predictions with observations by small “bird” radars at 2 locations to assess external validity. These approaches could provide rigorous evidence of the accuracy of the interpolation model. However, as presented, it is difficult for me to assess how informative and accurate this model is for continental monitoring of bird populations due to a lack of critical methodological details and results for assessing model fit. I provide more details about this below:
Section 2.5.2. The methods provided for how the bird radars operated and quantified bird migration density aloft is insufficient (only two sentences to describe the methods of this component!). I assume the methods of bird migration rate quantification are described in detail in the 2 citations provided, but a bit more information is really needed for this manuscript too, especially about how you compare the bird radar data to the model-interpolated predictions. Please explain how you converted migration traffic rates (a flow measure) from bird radars into a bird density measure. How large of a 2-dimensional area did the bird radars sample? Do they match the size of one or more grid cells of interpolated data from the geostatistical model? How did you spatially match the model-interpolated bird density to that measured by the bird radars? Where exactly were the two sampling locations in relation to the nearest weather radars? Where they within the range of observed weather radar (i.e., 25 km) used to characterize bird density, or within an area where bird migration was truly interpolated? This distinction is rather critical because the former would provide direct corroboration of weather radar and bird radar observations, while the latter would provide external validation of the geostatistical model predictions. I assume you are attempting the later. Did the sampling effort for the bird radars match that for the weather radar data sampling? In Lines 193 & 194 there are samples sizes of 164 and 264 given for the bird radars. What do these samples represent exactly? Is it bird migration intensity sampled within a 2 degree lat and long area over a 15 minute time period? Why do sample sizes differ between the sites?
Section 3.1.1 - You state that the “variances of the normalized error of estimation of each radar are close to 1” , which indicates good model accuracy. I don’t understand your interpretation of the normalized error variance. How does the magnitude of the normalized error inform you of the accuracy of the model? How are you “normalizing” the error exactly? Normalization usually means to scale a variable to have a values between 0 and 1. However, it could also mean "standardizing" the data to have a mean of zero and a standard deviation of 1. If you are standardizing the error in this way, than your "normalized variance" should be 1, regardless of how well the model fits the data. Perhaps you could instead provide a measure of the amount of variance explained by the model to the cross validation test data (e.g., R-squared)? Sorry, as a radar ornithologist, I am more familiar with r-squared as a measure of model fit than variance of normalized error. Perhaps you just need to explain your interpretation better for an ornithologist?
Section 3.1.2 – Again, I found it difficult to assess the goodness of fit of the model to external data from your statement that “Overall, the root-mean-square error of the non-transformed variable (i.e. the actual bird densities) was around 20 bird/km2 for both radars, which demonstrates the good performance of the model at these two test locations.” RMSE is an absolute error estimate that, on it’s own, is not an interpretable measure of the goodness of fit of a model, as far as I am aware. For that, you would need to convert RMSE into a relative error estimate such as through calculating an R-squared value. Please report the mean (or median) and variance of the observed bird densities at the bird radar sites and calculate an R-squared for the model’s goodness of fit to the bird radar observations. You present the mean and median bird densities in the supplemental (line 395) of the weather data. Based on this median bird density of 8 birds/km2, and RMSE of 20 birds/km2 seems really high (magnitude is >2x the median), which I would interpret as potentially a poorly fit model.
In general, a kriging model is rather generalized and, as you state in line 142, “leads to excessively smooth interpolation maps [e.g. 25] and thus fails to reproduce the fine-scale texture of the process at hand. Consequently, estimation should be complemented by geostatistical simulation”. However, it is not clear what is meant by “simulation”, nor how this provides finer-scale information than the kriging model predictions per se. Please provide methods describing how simulations were run. Sorry, I can’t find any description of the simulation methods anywhere. What aspects of the simulations were stochastic? Furthermore, there is no quantitative assessment of whether the stochastic simulations provide more accurate results compared to the mean predictions from the deterministic kriging model. Without such an assessment, I’m not sure the simulations add much to the manuscript. Therefore, I recommend dropping presentation of simulation results.
Line 226-228: There are results that should be reported in the results section.
Section 4.1: How do you justify relying solely on autocorrelation rather than also incorporating external drivers to inform predictions of bird migration intensity? You need to provide evidence (or citation to other studies) to support your claim that your interpolation method implicitly accounts for the influence of external drivers of bird migration. You also state that adding such external covariables to the interpolation model "is only advantageous if the correlation between bird densities and these covariates is stronger than the spatio-temporal correlation of the nearby radars." Again, you need to provide evidence to support this statement. I suggest that you test this statement by developing a co-kriging model that incorporates external drivers. For example, there IS an easily-accessible, continent-wide weather dataset (e.g., NCEP) that could provide covariates (rain, wind, temperature) to model bird migration intensity via a co-kriging model. Your statement would have strong support if you could show evidence of a lack of improved model fit of the co-kriging relative to your simple kriging model.
Lines 239 - 241: You present results from the supplemental material that should go in the results section. Furthermore, if these are important results, which I think they are, than don't sequester them to the supplemental document. They should be incorporated into the main manuscript. Please also provide support for using the "distance for which the auto-covariance
has reached 10% of its baseline value" as a threshold of "decorrelation". I am not familiar with this practice.
Appendix A: It is not clear from how many radars data were actually used for analysis. The methods mention 69 radars, but then the Appendix mentions that data from 11 radars were excluded for quality concerns and another 4 radars were excluded because of geographic isolation. So does that mean you only used data from 54 radars (69-11-4=54)? Or were the 69 radars, the number of radars left after data exclusion?
Figure C5 - What do the dotted horizontal lines in the plots depict? I am guessing it is the 10% baseline of covariance that you state indicates the limit of the autocorrelation in the data? If so please add that to the figure caption.
There are now hybrid machine learning models that can incorporate autocorrelation and perform better than kriging approaches. See https://www.sciencedirect.com/science/article/pii/S0341816218305332
You might mention these approaches in your discussion for future model development efforts.
Best, Jeff Buler
Round 2
Reviewer 1 Report
I appreciate the effort by the authors to standardize methodology and improve the presentation. I find the manuscript to be substantially improved. I was able to understand a number of aspects of the modeling that were fuzzy to me before, and the authors have provided good motivation for their overall modeling approach.
I have only minor comments on the revised manuscript:
Line 85. "stationary Gaussian RV" —> "stationary Gaussian process". A random variable cannot be stationary, but a random process (an infinite collection of random variables indexed by space/time) can be
Line 109. Awkward to start a sentence with notation. Possible rewording: "The parameters $0 < \alpha$, $\gamma < 1$ control regularity ..."
Eqs. (4,5), Line 114. I suspect the usage in general is not that consistent, but I believe the Dirac delta function is defined to have infinite magnitude at zero, whereas you want a function that has unit magnitude at zero. I believe "indicator function" or perhaps the Kronecker delta function would be a more appropriate term.
Eq. (12). The notation on the RHS of the definition of the quantile function is unusual to me. I would consider defining it as
\inf \{ b : \Pr(B(s_0, t_0) < b) \geq \rho}
I believe this is more standard.
Line 177 bird -> birds
Line 193-194 "For instance, a normalized error of 1 corresponds to the estimation value being one standard deviation above the measured value." Add “estimated” before standard deviation?
Lines 229–233. "The biases do not show any clear spatial pattern (Figure D2 in Appendix D) and therefore such biases do not originate from the geostatistical model itself (or of country-specific data quality). Rather, these radar-specific biases probably come from either the data itself, such as birds non-accounted for (e.g. flying below the radar), or errors in the cleaning procedure (e.g. ground scattering)". This conclusion seems too assertive and I don't fully understand the reasoning. I believe a bias is always a property of a model and not of data. I suspect what you mean is that the biases are not spatially auto-correlated, so you believe the model is correctly modeling the existing spatial autocorrelation structure. I would actually suggest deleting all the commentary and saying "The biases do not show any clear spatial pattern."
Fig 4. I believe the (a) and (b) markers are missing from the figure? I assume top panel is (a)
Fig 4. Lines 239–241 say "First, the estimations of the model correctly reproduce the night-to-night migration intensity, with the exception of a few nights (27-30 September for Herzeele and 26-27 September for Sempach)." I had a hard time seeing the visual evidence for this. Do you mean cases when the red dots are all well outside the uncertainty band? I see this for several days on the bottom plot, but not necessarily the ones you mentioned.
Line 258 adequatly —> adequately
Line 294 co-kriging. I still question whether co-kriging is the appropriate term here. The most standard way to model using co-variates would be to use them to model the mean function (i.e., “universal” or “regression” kriging). If you mean to model the covariates as other random processes, I suppose it would be co-kriging, but I don’t see the obvious motivation for that.
Lines 317–319. “a similar geostatistical approach can be used to interpolate flight speeds and directions that are also derived from weather radar data.” This reminds me of a closely related paper I think is relevant to your paper and should probably be discussed:
Rico Angell and Daniel Sheldon. Inferring Latent Velocities from Weather Radar Data using Gaussian Processes. Neural Information Processing Systems (NeurIPS), 2018.
Reviewer 3 Report
I am satisfied with the changes to the manuscript.
Author Response
Thank you for the helpful review.
Best wishes,